# Metabolomic profiles of sleep-disordered breathing are associated with hypertension and diabetes mellitus development

Ying Zhang [1], Bing Yu [2], Qibin Qi [3], Ali Azarbarzin[4], Han Chen[5], Neomi A. Shah[6], Alberto R. Ramos[7], Phyllis C. Zee[8], Jianwen Cai[9], Martha L. Daviglus[10], Eric Boerwinkle[2], Robert Kaplan[3,11], Peter Y. Liu [12], Susan Redline [4] & Tamar Sofer [4,13,14] ✉

Sleep-disordered breathing (SDB) is a prevalent disorder characterized by recurrent episodic upper airway obstruction. Using data from the Hispanic Community Health Study/Study of Latinos (HCHS/SOL), we apply principal component analysis (PCA) to seven SDB-related measures. We estimate the associations of the top two SDB PCs with serum levels of 617 metabolites, in both single-metabolite analysis, and a joint penalized regression analysis. The discovery analysis includes 3299 individuals, with validation in a separate dataset of 1522 individuals. Five metabolite associations with SDB PCs are discovered and replicated. SDB PC1, characterized by frequent respiratory events common in older and male adults, is associated with pregnanolone and progesterone-related sulfated metabolites. SDB PC2, characterized by short respiratory event length and self-reported restless sleep, enriched in young adults, is associated with sphingomyelins. Metabolite risk scores (MRSs), representing metabolite signatures associated with the two SDB PCs, are associated with 6-year incident hypertension and diabetes. These MRSs have the potential to serve as biomarkers for SDB, guiding risk stratification and treatment decisions.

Sleep-disordered breathing (SDB) is a common yet underdiagnosed disorder. It is estimated to affect approximately 17% and 34% of middle-aged female and middle-aged male individuals, respectively[1], but diagnosed in less than 15% of individuals with clinically significant disease[2,3]. SDB is characterized by recurring episodes of complete (apneas) or partial (hypopneas) upper airway obstruction, often accompanied by oxyhemoglobin desaturation and/or sleep fragmentation. Symptoms include snoring and excessive daytime sleepiness[4]. A growing body of epidemiological studies has found that SDB is associated with increased risks for vascular and metabolic diseases, including stroke, coronary heart disease, hypertension, and diabetes mellitus[5–9].

Underlying mechanisms proposed to associate SDB with the cardiometabolic conditions include: chronic hypoxemia, particularly nightly exposures to intermittent hypoxemia and re-oxygenation[10]; dysregulated proinflammatory responses[11]; increased oxidative stress[12], imbalanced gut microbiome[13], hormonal imbalance[14], among others. Emerging evidence has shown that intermittent hypoxemia, especially high frequency desaturations, modulates the inflammatory response differently from chronic sustained hypoxemia[15]. While recent work has examined specific aspects of SDB that best predict incident outcomes[16–18], only a few studies have tried to model more complex exposures by combining multiple SDB measures together[19,20]. Various SDB measures, such as the frequency of obstructive events (e.g.,

Respiratory Event Index (REI)), sleep-apnea specific hypoxic burden[21], minimum oxyhemoglobin saturation during sleep, apnea and hypopnea event duration[22], and others, while capturing different characteristics of SDB-related physiological stressors, tend to be correlated. Given the increasing recognition of the heterogeneity and complexity of SDB[23], indices that combine multiple measures of SDB by accounting for the correlation among them may provide powerful approaches for studying both SDB biology and risk stratification of incident cardiometabolic outcomes.

Metabolites, reflective of the products and intermediates of metabolism, can provide biomarkers useful for disease prediction and subtyping[24]. Studying SDB-associated metabolites may yield insights into the metabolic environment of the disorder, elucidate sex differences, and suggest SDB subtypes and related molecular mechanisms involved in the progression of cardiometabolic conditions. Untargeted metabolomic profiling is the comprehensive identification and quantification of small metabolite molecules within the biological system, and has begun to be used in sleep research to understand the cellular process such as sleep/wake regulation[25,26], as a window to the timing of peripheral molecular clocks and oscillators during circadian misalignment[27], and to detect biomarkers of sleep restriction[28] and neurological degeneration among patients with obstructive sleep apnea (OSA)[29]. In a recent study[30], we identified metabolites associated with moderate to severe OSA (defined as a REI >= 15) and constructed an index composed of 14 metabolites, associated with OSA cross-sectionally, in two independent datasets. Another recent study[31] identified metabolites associated with SDB and metabolites that changed levels following SDB treatment using continuous positive airway pressure, though without multiple testing correction. Further demonstrating the potential clinical utility of untargeted metabolite profiling, prediction models incorporating metabolites outperformed clinical predictors for some conditions[32]. Thus, untargeted metabolomics may provide a unique opportunity both for the development of biomarkers for SDB, and for utilizing such biomarkers for SDB-related risk stratification: identifying patients with increased risks for other chronic diseases.

We hypothesize that by combining SDB measures, and next, identifying and combining changes in their associated metabolomic environment, we can construct new SDB biomarkers that may offer additional utility compared to standard measures for identifying individuals at high risks for progression of cardiometabolic disease (Fig. 1). We use a data-driven, unsupervised principal component (PC) analysis to first construct two SDB summary measures based on several physiological phenotypes. We then study the association of the SDB PCs and the metabolic environment in a large population-based study with a high-dimensional set of measured metabolites using two methods: (1) association analysis of individual metabolites with each SDB PC, and (2) least absolute shrinkage and selection operator (LASSO) regression to identify a subset of metabolites that together best associate with SDB PCs. Based on LASSO-selected metabolites and their effect estimates, we develop SDB PC-specific metabolite risk scores (SDB-MRS). To validate our results, we use a discovery-replication approach where we separate datasets of individuals sampled from the same target population. We then study the SDB PC-specific MRS associations with incident hypertension and diabetes mellitus.

## Results

### Metabolomics sample characteristics
The main, batch 1, discovery dataset (used for SDB SMA analysis and LASSO regression) included 1874 female participants (mean age = 42.8), and 1425 male participants (mean age = 41.6), and the replication dataset included 960 female participants (mean age = 51.9) and 562 male participants (mean age = 51.2) from batch 2 (Table 1 and Supplementary Data 1). Consistent with their older age, the prevalence of moderate to severe SDB was higher in batch 2 compared to batch 1

participants (REI3 ≥ 15, 13.8% compared to 11.5% in batch 1 participants); similarly, comorbidities were higher in batch 2 participants (30.1% prevalent diabetes mellitus and 45.7% prevalent hypertension, compared to 20.4% and 32.2%, respectively, in batch 1).

### SDB PC1 and SDB PC2 characterize study population on different dimensions
In total, 11,653 HCHS/SOL study participants with complete SDB measures were included in the principal component analysis of SDB phenotypes. Supplementary Data 2 shows the sample characteristics stratified by sex while accounting for sampling weights, so that means and proportions are representative of the HCHS/SOL target population. The first two principal components of the SDB measures accounted for 79.8% of the total variance (SDB PC1: 65%, SDB PC2: 14.5%; Supplementary Fig. S1). For both PCs, higher values indicate more severe hypoxemia. However, PC1 is also characterized by more frequent respiratory events while PC2 is characterized by shorter respiratory events. Specifically, high SDB PC1 is correlated with increased REI3 (Spearman correlation coefficient $\rho = 0.67$) and REI0 ($\rho = 0.77$), increased hypoxic burden ($\rho = 0.67$), high percentage of sleep time with SpO2 < 90% ($\rho = 0.45$), decreased average oxygen saturation ($\rho = -0.64$) and lower minimum oxygen saturation ($\rho = -0.79$). High SDB PC2 is mostly correlated with reduced average event length ($\rho = -0.53$), lower average ($\rho = -0.38$) and minimum oxygen saturation ($\rho = -0.32$), and increased percentage of sleep time with SpO2 < 90% ($\rho = 0.2$) (Fig. 2).

To better understand the phenotypic characteristics that SDB PC1 and PC2 represent, we also compared the populations defined by the top and bottom 10% of SDB PC1 and PC2 (Table 2 and Supplementary Data 3). The top 10% compared with the bottom 10% SDB PC1 was comprised of individuals who were on average older and have a higher BMI; more likely to be male and have prevalent and incident hypertension and diabetes mellitus (diabetes henceforth); and more likely to have history of smoking. The top 10% SDB PC2 compared to the bottom PC2 included participants who were slightly younger, less likely to be males, and more likely to be current smokers but did not differ in rates of baseline and incident hypertension and diabetes (Table 2 and Supplementary Data 3). As for sleep disturbance traits, the top and bottom 10% SDB PC1 participants reported similar insomnia symptoms according to the Women's Health Initiative Insomnia Rating Scale and similar sleep quality (typical night's sleep in the past 4 weeks being restless or very restless), but reported more severe excessive sleepiness, more frequent snoring and shorter sleep duration, whereas the top 10% SDB PC2 participants reported worse insomnia symptoms and sleep quality compared to the bottom 10% SDB PC2.

### Single metabolite associations (SMA) with SDB PCs
Figure 3 shows 15 metabolites associated with SDB PC1 and 4 metabolites associated with SDB PC2 (FDR-corrected $p < 0.05$) in HCHS/SOL batch 1 in Model 1 (the corresponding effect estimates, and biochemical information are provided in Supplementary Data 4 and 5). Among the 15 SDB PC1 metabolites, two metabolites, pregnanolone/allopregnanolone sulfate and glucuronide of C10H18O2 (8), had replicated associations (FDR-corrected one-sided $p < 0.05$) in batch 2 in Model 1 analysis (Fig. 4), and remained associated (pregnanolone/allopregnanolone sulfate: FDR-corrected one-sided $p = 0.036$ in model 2, FDR-corrected one-sided $p = 0.039$ in model 3; glucuronide of C10H18O2 (8): FDR-corrected one-sided $p = 0.030$ in model 2, FDR-corrected one-sided $p = 0.032$ in model 3) with PC1 when adjusted for additional lifestyle and comorbidity covariates in batch 2. Three of the four metabolite associations with SDB PC2 in batch 1 replicated in batch 2 (FDR-corrected one-sided $p < 0.05$) in Model 1 and 2, all of which were sphingomyelin lipids - sphingomyelin(d18:2/24:2), sphingomyelin(d18:2/24:1,d18:1/24:2), and sphingomyelin(d18:2/23:0,d18:1/

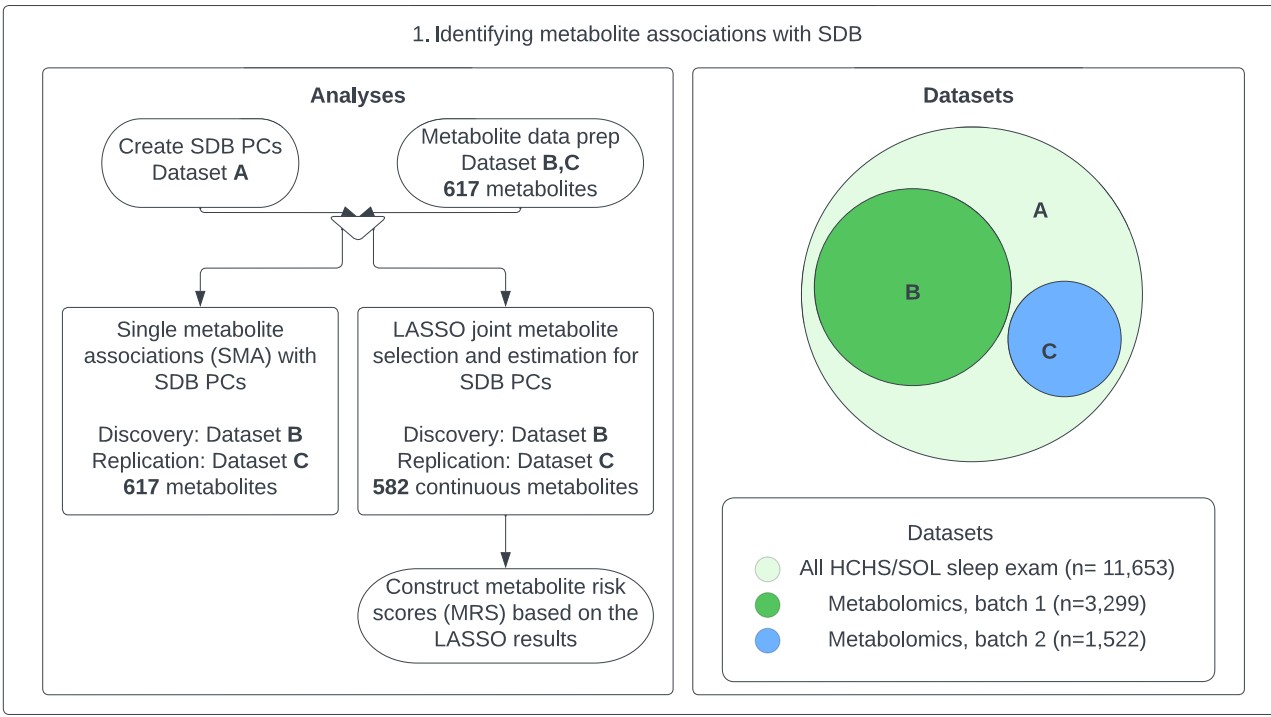

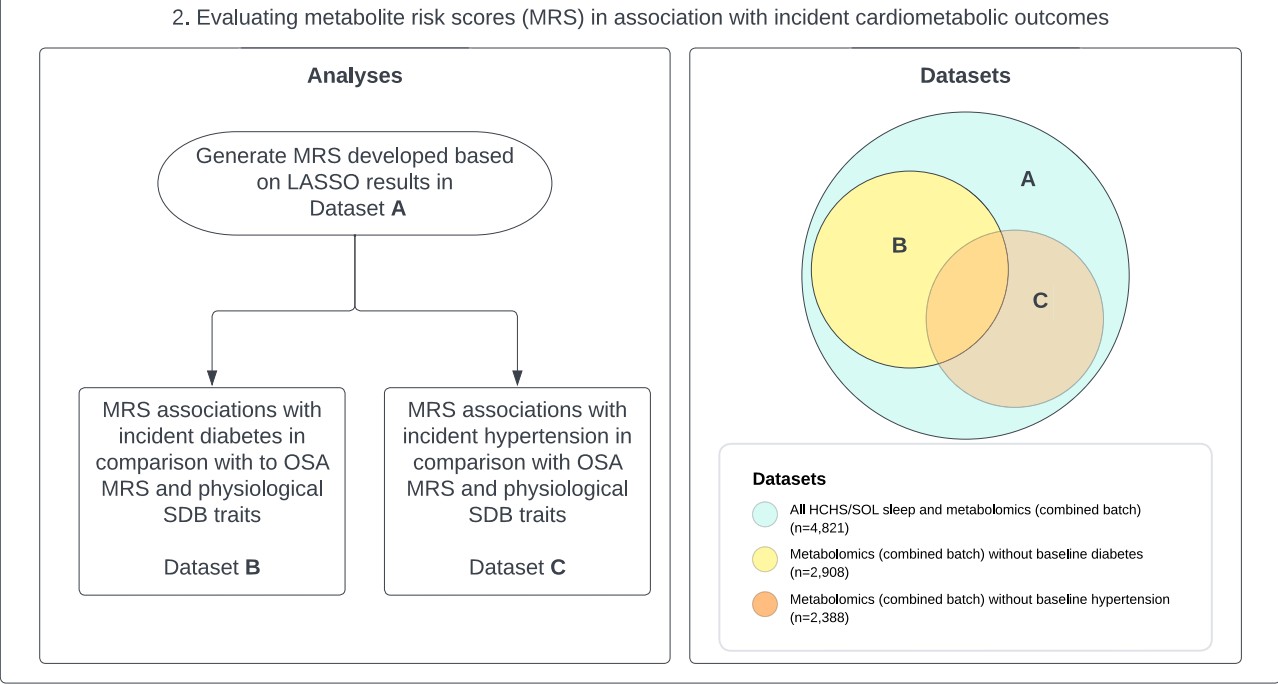

**Fig. 1 | Study design diagram.** SDB sleep disordered breathing, PC principal component, LASSO least absolute shrinkage and selection operator, OSA MRS metabolite risk score calculated based on coefficients from LASSO regression trained to predict OSA in previous publication[30], HCHS/SOL the Hispanic Community Health Study/Study of Latinos. Baseline diabetes are based on American Diabetes Association definition[69], defined as fasting glucose >=126 mg/dL, or post-OGTT glucose >=200 mg/dL or A1C > = 6.5%, or self-report of diabetes; baseline hypertension is defined as systolic or diastolic BP greater than or equal to 140/90, respectively, or current use of antihypertensive medications.

23:1, d17:1/24:1). Full results from the SMA sex-combined analysis are provided in Supplementary Data 6.

In the sex-specific SMA, tauro-beta-muricholate, a lipid from the bile acid metabolism pathway, was associated with SDB PC1 (FDR-corrected $p < 0.05$) among males, while no metabolite was identified for SDB PC2 in male-only analysis after FDR correction. The association of tauro-beta-muricholate with SDB-PC1 in males did not replicate in batch 2 (Supplementary Data 7). In female-specific discovery analysis, ten metabolites were associated with SDB-PC1, of which eight were discovered in the sex-combined SMA analysis, and two, 3-hydroxyoctanoylcarnitine (1) and 3-hydroxyoctanoylcarnitine (2), were unique to the sex-stratified analysis. A single metabolite, allantoin, was associated with SDB-PC2 among females (Supplementary Data 6). Among the twelve metabolites identified in either the

**Table 1 | Characteristics of Hispanics/Latinos represented by the metabolomic analytic sample from the HCHS/SOL study**

| Mean (SD)[a] | Batch 1 | | | Batch 2 | | |
|---|---|---|---|---|---|---|
| | Female | Male | Overall | Female | Male | Overall |
| n | 1874 | 1425 | 3299 | 960 | 562 | 1522 |
| Age at baseline | 42.82 (15.10) | 41.62 (14.94) | 42.22 (15.03) | 51.89 (12.31) | 51.18 (13.73) | 51.57 (12.96) |
| BMI (kg/m$^2$) | 30.28 (6.86) | 28.87 (5.37) | 29.58 (6.20) | 30.22 (5.96) | 28.68 (5.04) | 29.54 (5.62) |
| Current alcohol drinking | 733 (39.1) | 886 (62.2) | 1619 (49.1) | 340 (35.5) | 313 (55.7) | 653 (43.0) |
| Current smoker | 287 (15.3) | 382 (26.8) | 669 (20.3) | 145 (15.1) | 142 (25.4) | 287 (18.9) |
| Physical activity (MET-min/day) | 481.45 (757.57) | 943.14 (1197.58) | 711.22 (1027.14) | 331.97 (595.88) | 798.99 (1208.27) | 538.38 (946.79) |
| The Alternate Healthy Eating Index (AHEI 2010) | 46.81 (7.47) | 48.87 (7.47) | 47.83 (7.54) | 48.31 (7.03) | 50.07 (7.42) | 49.09 (7.25) |
| OSA status = OSA, REI3 ≥ 15 (%) | 141 (7.5) | 240 (16.8) | 381 (11.5) | 94 (9.8) | 116 (20.6) | 210 (13.8) |
| REI0 (events/hr) | 14.24 (16.78) | 22.20 (21.53) | 18.21 (19.69) | 18.87 (17.42) | 26.05 (21.76) | 22.05 (19.78) |
| REI3 (events/hr) | 3.82 (7.73) | 8.26 (15.09) | 6.03 (12.18) | 5.59 (10.28) | 10.03 (15.17) | 7.56 (12.87) |
| Average length of each respiratory event (seconds) | 17.82 (4.16) | 19.61 (4.52) | 18.71 (4.43) | 18.47 (4.71) | 21.06 (5.31) | 19.62 (5.14) |
| Percentage sleep time with SpO2 < 90% | 0.40 (1.50) | 1.10 (4.23) | 0.75 (3.18) | 0.67 (2.26) | 1.48 (4.13) | 1.03 (3.25) |
| Sleep-related time in hypoxia (5% sleep <90% saturation) (%) | 42 (2.2) | 82 (5.8) | 124 (3.8) | 32 (3.3) | 36 (6.4) | 68 (4.5) |
| Hypoxic burden (%minute/hour) | 14.17 (23.38) | 26.62 (44.70) | 20.37 (36.17) | 20.41 (30.99) | 33.59 (47.14) | 26.26 (39.51) |
| Minimum SpO2% | 88.27 (5.09) | 86.70 (6.18) | 87.49 (5.71) | 86.93 (6.11) | 85.38 (6.80) | 86.25 (6.47) |
| Average SpO2% | 96.66 (0.68) | 96.38 (1.09) | 96.52 (0.92) | 96.43 (0.82) | 96.19 (1.01) | 96.32 (0.91) |
| Baseline diabetes status (ADA)[b] (%) | 381 (20.3) | 292 (20.5) | 673 (20.4) | 277 (28.9) | 181 (32.2) | 458 (30.1) |
| Baseline hypertension status[c] (%) | 613 (32.7) | 449 (31.5) | 1062 (32.2) | 439 (45.7) | 256 (45.6) | 695 (45.7) |
| Incident diabetes (ADA)[b] (%) | 183 (12.9) | 122 (12.8) | 305 (12.9) | 118 (12.6) | 80 (14.7) | 198 (13.4) |
| Incident hypertension[c] (%) | 172 (9.2) | 127 (8.9) | 299 (9.1) | 137 (14.3) | 85 (15.1) | 222 (14.6) |

[a]Means and percentages were weighted using sampling weights to provides estimates of the HCHS/SOL target population characteristics.

[b]Baseline and incident diabetes are based on American Diabetes Association definition[69], defined as fasting glucose >=126 mg/dL, or post-OGTT glucose >=200 mg/dL or A1C > = 6.5%, or self-report of diabetes.

[c]Baseline and incident hypertension is defined as systolic or diastolic blood pressure greater than or equal to 140/90 respectively, or current use of antihypertensive medications.

male- and female-specific SMA analysis, only the associations of pregnanolone/allopregnanolone sulfate and glucuronide of C10H18O2 (8) with PC1 were close to replicated in batch 2 among females (one sided-$p < 0.05$, Supplementary Data 7). When tested for evidence of interaction with sex, only tauro-beta-muricholate had statistically significant interaction effect (FDR-corrected $p = 0.014$) (Supplementary Data 8).

Given that half of the discovered and replicated SDB PC1 metabolites were from the progesterone steroids biosynthesis pathway, we compared and visualized the concentration levels of the eight progesterone steroids sulfate metabolites with statistically significant associations with SDB PC1 after FDR correction in batch 1 by age groups in each sex stratum. As age increases, we observed a decreasing trend in the levels of circulating progesterone steroids sulfate metabolites in both men and women. The patterns become more visible in the rank-normalized metabolites (Supplementary Fig. S2). Sulfated metabolites of progesterone − 5alpha-pregnan-3beta,20alpha-diol disulfate, 5alpha-pregnan-3beta,20alpha-diol monosulfate (2), and 5alpha-pregnan-3beta,20beta-diol monosulfate (1), 5alpha-pregnan-diol disulfate and Pregnanolone/allopregnanolone sulfate, were higher among younger women compared to younger men (in age groups <40 and 40−45), while the differences diminished in older age groups (50−55, 55−60, and >60) that would typically include post-menopausal women. The circulating pregnenolone steroids sulfate metabolites pregnanediol sulfate (C21H34O5S)*, pregnenetriol sulfate*, and pregnenolone sulfate, were higher in men compared to women across all age groups. The patterns were similar in the two batches.

To compare the SDB SMA results with our prior publication reporting OSA-metabolite associations[30], we examined the overlap between the OSA and SDB PCs metabolite associations. While none of the metabolites associated with SDB PCs were included in the OSA

SMA analysis, all metabolites reported as associated with OSA were included in the present SDB SMA analysis (Supplementary Data 9). Briefly, all metabolites had some evidence of association with SDB PC1 ($p < 0.1$ in either batch), but only two metabolites had FDR-corrected association $p < 0.05$ in the batch 1 discovery analysis.

In a secondary analysis, we compared the associations between SDB PCs-associated metabolites (FDR-corrected $p$-value based on batch 1 analysis for each SBP PC1 or PC2; metabolites reported in Fig. 3) and the 7 individual SDB phenotypes comprising the SDB PCs. The four SDB PC2 associated metabolites had the largest evidence of association with average oxyhemoglobin saturation during sleep and with REI3. SDB PC1 associated metabolites were marginally associated with multiple SDB phenotypes, i.e., did not seem to strongly reflect associations with any specific individual SDB phenotype. Three metabolites, 1-stearoyl-2-arachidonoyl-GPC (18:0/20:4), 2-linoleoylglycerol (18:2) and glucuronide of C10H18O2 (8), had low evidence of associations with all SDB phenotypes when evaluated individually ($p > 0.01$; Supplementary Fig. S3), highlighting the contribution of the PCA approach.

### LASSO regression for joint selection and estimation of metabolite associations with SDB PCs in HCHS/SOL batch 1

To identify a set of metabolites that were jointly associated with SDB PCs, we also implemented a LASSO regression in HCHS/SOL batch 1 (discovery dataset), both in sex-combined and sex-stratified study samples. 112 metabolites were identified for SDB PC1, and 57 metabolites for SDB PC2, with 14 metabolites overlapping between the two groups. The breakdown of super pathways of the metabolites are shown in Supplementary Fig. S4 and coefficients for all metabolites from LASSO trained in sex-combined and sex-stratified samples are provided in Supplementary Data 10.

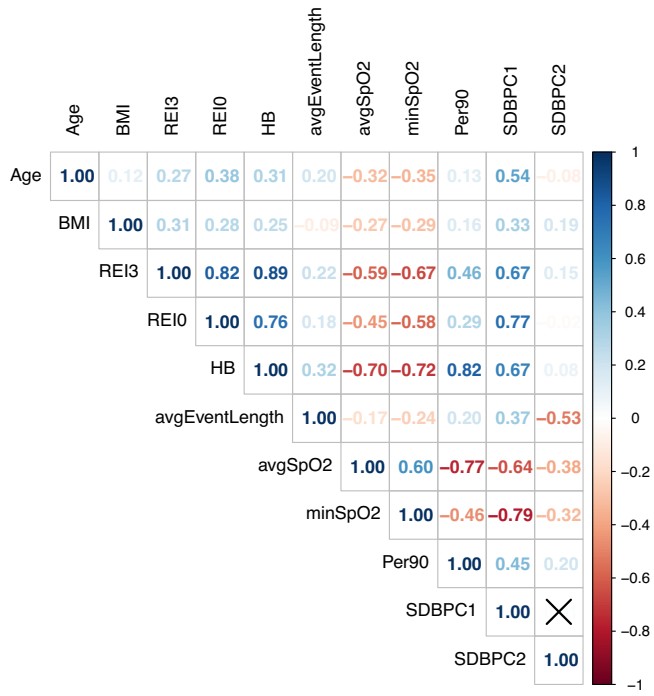

**Fig. 2 | Estimated correlations between age, BMI, SDB phenotypes and SDB PCs.** REI3: Respiratory Event Index (REI) computed over all respiratory events, defined as apneas or hypopneas with at least 50% cannula flow reduction for a minimum duration of 10 s with >=3% oxygen desaturation; REI0 REI computed over all respiratory events regardless of oxygen desaturation; HB hypoxic burden; avgEventLength the average length of apnea and hypopnea events (combined); minSpO2 minimum oxyhemoglobin saturation during sleep; avgSpO2 average oxyhemoglobin saturation during sleep; Per90 percentage of sleep time with oxyhemoglobin saturation below 90%; SDBPC1 the first principal component of the seven sleep disordered breathing traits included in the principal component analysis; SDBPC2 the second principal component of the seven sleep disordered breathing traits included in the principal component analysis. Source data are provided in Supplementary Data 17.

We constructed SDB PC1 MRS and SDB PC2 MRS for batch 1 and batch 2 HCHS/SOL participants based on results from the LASSO penalized regression. Study sample means and SD used in standardizing the MRSs are provided in Supplementary Data 11. As expected by construction, all SDB MRSs were significantly associated with their corresponding SDB PCs in batch 1 in all models. The associations replicated for sex-combined SDB MRSs but not for sex-specific SDB PC 2 MRS for females in batch 2 (Table 3). The sex-specific SDB PC MRSs, when replicated in batch 2, did not show stronger associations with their corresponding SDB PCs.

In the sensitivity analysis, we constructed SDB PC MRSs in a process restricted exclusively to batch 1 participants (including both PCA and MRS derivation). We then constructed the resulting b1-SDB PCs and b1-SDB PC MRSs among batch 2 participants. In batch 2, b1-SDB PCs and their corresponding b1-SDB PC MRSs were highly associated (Supplementary Data 12).

## Associations with incident cardiometabolic outcomes

In the HCHS/SOL sleep study target population, SDB PC1 showed positive associations with incident diabetes and hypertension over an average of 6.1 years (4.3–9.4 years) in both Model 1 and 2 among the individuals without diabetes or hypertension at baseline, respectively. These composite phenotypes showed stronger associations than the individual SDB measures REI3 and hypoxic burden. SDB PC2 was not significantly associated with either incident outcome (Supplementary Data 13).

In the batch-combined analysis, both SDB MRSs were significantly associated with increased incidence rate ratio (IRR) for incident hypertension and incident diabetes mellitus, when adjusted for demographic and lifestyle risk factors (Fig. 5 and Supplementary Data 14). One SD increase of SDB PC2 MRS was associated with a 28% (IRR: 1.28 95% CI: 1.12–1.46, $p = 0.0004$) higher incidence rate of hypertension and a 30% (IRR: 1.30 95% CI: 1.12–1.51, $p = 0.0005$) higher incidence rate of diabetes, adjusted for demographic and lifestyle covariates (Supplementary Data 14). The effect estimates were slightly lower for SDB PC1 MRS when adjusting for the same covariates (Fig. 5 and Supplementary Data 14). For comparison, we also computed OSA MRS, 1 SD increase of OSA MRS was associated with a 42% (IRR: 1.42 95% CI: 1.27–1.60, $p < 0.0001$) higher incidence rate of hypertension and a 57% (IRR: 1.57 95% CI: 1.38–1.80, $p < 0.0001$) higher incidence rate of diabetes. None of the single metric physiological phenotypes (i.e., REI3, HB) were significantly associated with incident cardiometabolic outcomes in both models (Fig. 5 and Supplementary Data 14).

Secondary analysis was carried out by stratifying the study samples for incident diabetes into two subgroups: individuals with normal glucose regulation ($n = 1376$) and with impaired glucose regulation ($n = 1532$) at baseline. The observed associations between SDB PC1 MRS and incident diabetes became weaker in the two strata and lost statistical significance. The association between SDB PC2 MRS was statistically significant in both groups, with a stronger association observed in the normal glycemic group (IRR = 1.43 95% CI: 1.09–1.87, $p = 0.009$) compared to the impaired glucose regulation group (IRR = 1.25 95% CI: 1.08–1.44, $p = 0.002$), both adjusted for demographic and lifestyle covariates (Supplementary Data 14). The association between OSA MRS and incident diabetes also became weaker in both strata compared to the combined sample, and remained statistically significant only in the pre-diabetic group (IRR = 1.57 in the combined group, IRR = 1.38 in the pre-diabetic group, and IRR = 1.32 in the normal glycemic group) when adjusted for demographic and lifestyle covariates.

Focusing on OSA MRS, which had the strongest associations with incident outcomes among all MRSs, we also compared risks for incident outcomes by quartiles. Compared with the lowest quartile of the OSA MRS, the top quartile showed more than a three-fold increase in incidence rate for diabetes (IRR: 3.35 95% CI: 2.3–4.89, $p < 0.0001$) in Model 1 and its association remained statistically significant when adjusted for lifestyle covariates in addition (Model 2 IRR: 3.26 95% CI: 2.22–4.79, $p < 0.0001$) (Supplementary Fig. S5 and Supplementary Data 15).

There is no evidence supporting stronger associations with incident outcomes among SDB PCs MRSs trained in each sex stratum separately versus the sex-combined MRSs.

In the sensitivity analysis, we considered b1-SDB PC MRSs developed via an analytic pipeline including both PCA and LASSO analysis restricted to batch 1 participants who were not taking any antihypertensive or antidiabetic medications at baseline. We refer to these MRSs as b1-SDB PC1 and b1-SDB PC2 MRSs. These MRSs were each associated with increased risks for incident diabetes among batch 2 participants, while b1-SDB PC1 MRS but not b1-SDB PC2 was associated with increased risks for incident hypertension (associations were in both regression Models 1 and 2; Supplementary Data 16). In comparison, SDB PC2 MRS from the main analysis was associated with incident hypertension. Otherwise, the effect estimates of b1-SDB PC MRSs were generally stronger than the SDB PC MRSs developed in the primary analysis. Lack of an observed association of b1-SDB PC2 MRS with incident hypertension, and differences in the magnitude of effect sizes between the main and the sensitivity analyses should be interpreted with caution due to lower sample sizes in the sensitivity analysis and the difference in the sensitivity analysis sample representing healthier individuals.

**Table 2 | Characteristics of study participants with low and high values of SDB PCs**

| Mean (SD)[a] | SDB PC1 | | SDB PC2 | |
|---|---|---|---|---|
| | Top 10% | Bottom 10% | Top 10% | Bottom 10% |
| *n* | 1165 | 1166 | 1165 | 1166 |
| **Demographic variables** | | | | |
| Age at baseline | 53.92 (12.22) | 29.88 (10.89) | 38.22 (15.30) | 43.96 (14.38) |
| Sex = Male (%) | 649 (55.7) | 301 (25.8) | 404 (34.7) | 447 (38.3) |
| **Sleep disordered breathing** | | | | |
| REI0 (events/hr) | 57.79 (25.18) | 2.03 (1.67) | 14.70 (26.05) | 17.30 (13.22) |
| REI3 (events/hr) | 37.38 (23.05) | 0.08 (0.15) | 7.75 (19.34) | 2.17 (3.13) |
| Minimum SpO2% | 74.12 (7.34) | 91.92 (1.53) | 85.62 (7.58) | 91.49 (1.81) |
| Average SpO2% | 94.61 (1.65) | 97.07 (0.33) | 95.92 (1.49) | 97.02 (0.29) |
| Percent sleep time with SpO2 < 90% | 7.50 (8.65) | 0.01 (0.05) | 1.76 (5.87) | 0.02 (0.09) |
| Sleep-related time in hypoxia (5% sleep <90% saturation) (%) | 466 (40.0) | 0 (0.0) | 118 (10.1) | 0 (0.0) |
| Average length of each respiratory event (seconds) | 22.44 (5.32) | 16.03 (5.05) | 14.59 (2.73) | 23.38 (4.47) |
| Hypoxic burden (%minute/hour) | 109.42 (73.75) | 1.16 (1.11) | 20.74 (52.43) | 14.34 (11.73) |
| **Lifestyle variables** | | | | |
| BMI (kg/m²) | 33.94 (6.41) | 26.82 (5.74) | 30.78 (7.43) | 27.31 (4.79) |
| Current alcohol drinking | 536 (46.0) | 591 (50.7) | 551 (47.4) | 520 (44.7) |
| Current smoker | 176 (15.1) | 208 (17.9) | 311 (26.7) | 171 (14.7) |
| **Comorbidities** | | | | |
| Incident diabetes (ADA)[b](%) | 160 (18.6) | 46 (6.3) | 95 (12.0) | 118 (13.1) |
| Baseline diabetes status (ADA)[b](%) | 450 (38.6) | 86 (7.4) | 255 (21.9) | 195 (16.7) |
| Incident hypertension[c](%) | 125 (10.7) | 56 (4.8) | 107 (9.2) | 105 (9.0) |
| Baseline hypertension status[c](%) | 662 (56.8) | 100 (8.6) | 337 (28.9) | 317 (27.2) |
| **Self-reported sleep duration and sleep disturbance** | | | | |
| Sleep duration (hours) | 7.87 (1.42) | 8.32 (1.44) | 8.07 (1.57) | 7.89 (1.22) |
| Women's Health Initiative Insomnia Rating Scale total score | 6.95 (5.42) | 6.31 (5.22) | 7.38 (5.44) | 6.24 (5.15) |
| Typical night's sleep in past 4 weeks (restless or very restless) (%) | 213 (20.4) | 208 (20.7) | 256 (24.7) | 187 (18.2) |
| Take sleeping pills (%) | 83 (7.2) | 52 (4.5) | 115 (10.1) | 76 (6.6) |
| Trouble getting back to sleep (3 or more times a week) (%) | 214 (18.8) | 193 (17.0) | 252 (22.5) | 214 (18.9) |
| Wake up earlier than you plan (3 or more times a week) (%) | 269 (23.3) | 235 (20.5) | 259 (22.7) | 256 (22.2) |
| Wake up several times at night (3 or more times a week) (%) | 487 (42.2) | 328 (28.6) | 426 (37.3) | 383 (33.1) |
| Trouble falling asleep (3 or more times a week) (%) | 248 (21.5) | 262 (22.9) | 334 (29.3) | 263 (22.8) |
| Epworth Sleepiness Scale total score | 7.05 (5.31) | 5.25 (4.06) | 5.55 (4.54) | 5.52 (4.71) |
| Self-reported snoring (6-7 nights a week) (%) | 626 (65.7) | 67 (8.5) | 248 (29.1) | 200 (25.2) |

*SDB* sleep disordered breathing, *PC* principal component, *OSA* moderate or severe OSA (REI3 ≥ 15)

[a]Means and percentages were weighted using sampling weights to provides estimates of the HCHS/SOL target population characteristics.

[b]Baseline and incident diabetes are based on American Diabetes Association definition[69], defined as fasting glucose >=126 mg/dL, or post-OGTT glucose >=200 mg/dL or A1C > = 6.5%, or self-report of diabetes.

[c]Baseline and incident hypertension is defined as systolic or diastolic blood pressure greater than or equal to 140/90 respectively, or current use of antihypertensive medications.

## Discussion

We constructed new SDB measures based on seven correlated SDB phenotypes using PCA, weighted to represent the target population of the HCHS/SOL study. High scores for SDB PC1 appeared to characterize a SDB phenotype described by a high frequency of obstructive events and marked hypoxemia – a pattern typical of severe SDB and more often observed in men compared to women. In contrast, high SDB PC2 reflected a subphenotype that correlated mostly strongly with shorter event duration, and to a lesser degree, with hypoxia measures, while being almost uncorrelated with traditional event frequency measures (REI0 and REI3). In the HCHS/SOL, higher SDB PC2 was more common in younger women, individuals with more severe insomnia, self-reported poor sleep, frequent awakenings and longer sleep duration. SDB PC2 is highly correlated with shorter respiratory event duration, which has been reported in other cohorts to be more common in females, in younger individuals, and associated with higher arousal responses for any given change in oxygen saturation[33]. Moreover, in a discovery-replication approach within distinct subsamples

from the HCHS/SOL, we identified multiple metabolites individually associated with each SDB PC, as well as metabolites that are collectively associated with SDB. We used the latter set of metabolites to construct MRSs of SDB aggregating multiple metabolites. The SDB MRSs have stronger associations with incident cardiometabolic outcomes – diabetes mellitus and hypertension – compared to single SDB metrics, REI3 and hypoxic burden.

Higher concentrations of multiple sulfated metabolites of progesterone and its precursor pregnenolone were associated with lower (healthier) values of SDB PC1 (FDR-corrected $p < 0.05$) in the discovery dataset: pregnanolone/allopregnanolone sulfate (which replicated), as well as additional seven progestin steroids (highlighted in green in Fig. 6). Since progesterone in circulation is quickly metabolized by the liver and has a half-life of approximately 5 min[34], only the glucuronide and sulfate metabolites of progesterone steroids were measured in the Metabolon platform. Progesterone is a female reproductive hormone that is mostly synthesized in ovaries and by the placenta during pregnancy, and to a lesser degree in adrenal cortex and other

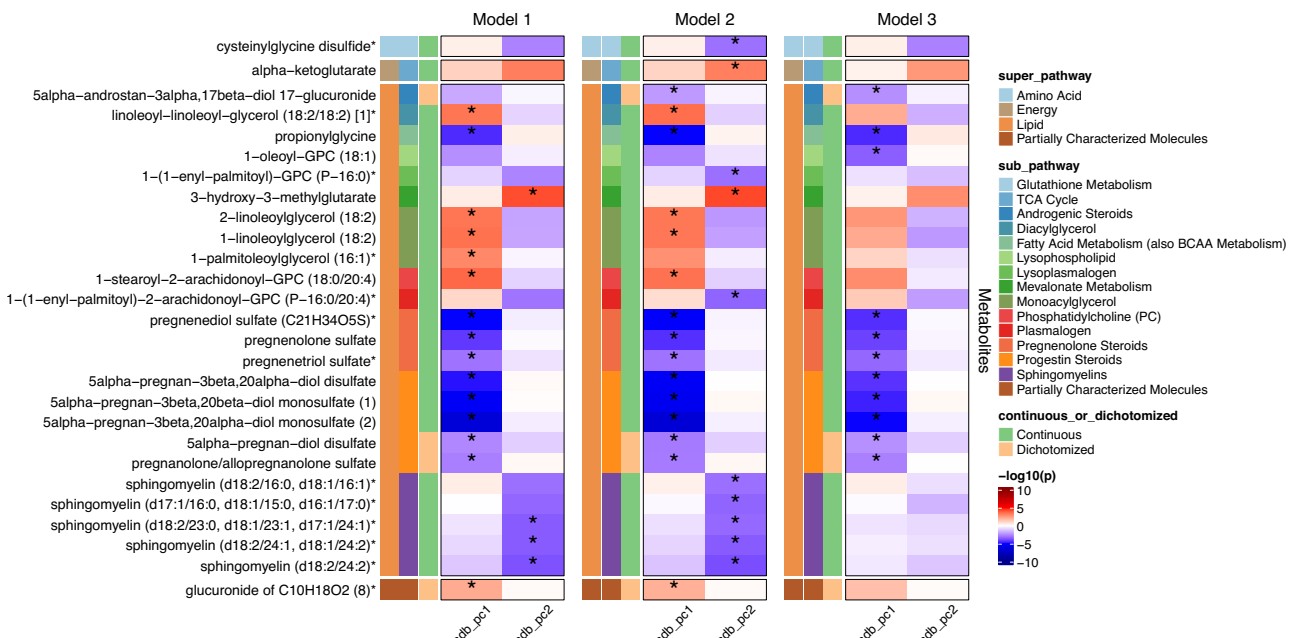

**Fig. 3 | Single metabolite association analysis in sex-combined analysis in batch 1.** -log10(*p*) is based on unadjusted two-sided *p* derived by accounting for the complex sampling design, using adjusted standard errors to compute the t-statistic in single metabolite association analysis with SDB PCs as dependent variables. *indicates FDR-corrected *p*-value, derived using the Benjamini-Hochberg method to control false discovery rate (FDR) for multiple testing among metabolites in all models for each SDB PC in the discovery dataset (batch 1) is below 0.05. sdb_pc1: the first principal component of the seven sleep disordered breathing traits included in the principal component analysis; sdb_pc2: the second principal component of the seven sleep disordered breathing traits included in the principal component analysis. Model 1 adjusted for demographic variables, including age, sex, field center, Hispanic/Latino background (Mexican, Puerto Rican, Cuban,

Central American, Dominican, and South American and other/multi) and body mass index (BMI). Model 2 adjusted for all model 1 covariates and lifestyle variables – alcohol use, cigarette use, total physical activity (MET-min/day), and diet (Alternative Healthy Eating Index 2010) in addition to demographic variables. Model 3 adjusted for all model 1 and 2 covariates and comorbidities - indicators for diabetes and hypertension, and continuous measures of fasting insulin, fasting glucose, HOMA-IR, HDL, LDL, total cholesterol, triglycerides, systolic blood pressure and diastolic blood pressure. Metabolites with *indicate they were identified based on accurate mass data, retention time and mass spectrometry but not reference standards. Therefore, the verification is not as robust as metabolites without *. Source data are provided in Supplementary Data 4.

tissues in both men and women, and in testes in men[35]. All progesterone steroid sulfates were present in both men and women in our dataset (Supplementary Fig. S2). The pattern of differences in these metabolites between sexes according to age suggests that sulfated metabolites of progesterone in women are of gonadal origin, whereas the sulfated metabolites of pregnenolone are of adrenal origin. Future studies will need to verify this possibility in cohorts where the date of menopause is known. If true, these data point to the possibility that different classes of steroids of different origins may be involved in the development of SDB, and its association with incident hypertension and diabetes mellitus.

The influence of progesterone- and pregnenolone-derived steroids on SDB has been a source of interest for decades given the considerable sexual dimorphism of this trait – i.e., the prevalence, severity, and physiological subtype all vary by sex. For example, while men are 3- to 4-fold more likely to have SDB than women, this sex differences attenuates after women reach menopause[36]. Women with SDB have a less collapsible airway, more hypopneas relative to apneas, and shorter event duration than men[33]. Progesterone is a proposed mechanism for protecting women from SDB. It is an anti-oxidant[37] that also is a respiratory stimulant that increases hypoxic and hypercapnic ventilatory response (including through effects on CO2 receptors), increases genioglossus muscle tone and decrease upper airway collapsibility[38–40]. Animal studies have shown the important roles of nuclear and membrane progesterone receptors mediating the stability of the breathing pattern and therapeutic effects in treating apnea of prematurity in both male and female mice[41]. SDB increases substantially among postmenopausal women[42–44], which may, at least in part, relate to changes in sex hormones. Two small cross-sectional studies reported

inverse between progesterone levels and OSA[43,45]. Post-menopausal women who use hormone replacement therapy that includes both estrogen and progesterone have lower respiratory event frequencies than their counterparts who do not use this therapy[46]. On the other hand, clinically induced sex hormone deficiency in young women has not associated with increased SDB[47]. The complexity of interpreting effects due to exogenous versus endogenous progesterone levels, bioavailability, receptor sensitivity, and the effects of other sex steroids has limited our understanding of role of progesterone steroids in the pathogenesis of SDB. In our study, the association with SDB PC1 suggests protective associations of progesterone steroids sulfate metabolites with SDB phenotype characterized by a high frequency of obstructive events and marked hypoxemia; while this association was observed in both women and men, this phenotype was more severe in men. Future longitudinal assessments of sex hormones and SDB may further elucidate mechanisms for SDB development across the life course.

Higher values of SDB PC2 were associated with lower concentrations of three sphingomyelins: sphingomyelin(d18:2/24:2), sphingomyelin(d18:2/24:1,d18:1/24:2), sphingomyelin(d18:2/23:0,d18:1/23:1, d17:1/24:1). Sphingomyelin has long been regarded as an inert structural component of the plasma membrane. However, recent studies have showed that it also plays an important role in the pathogenesis of cardiovascular, metabolic and neurodegenerative disease, potentially via mitochondrial dysfunction and abnormal reactive oxygen species (ROS) formation[48–51]. Dysregulation of sphingomyelin has been implicated in immune regulation, inflammation and apoptosis and acute and chronic lung pathology[52]. Several studies also reported circulating and urinary sphingolipids were altered among SDB patients and the

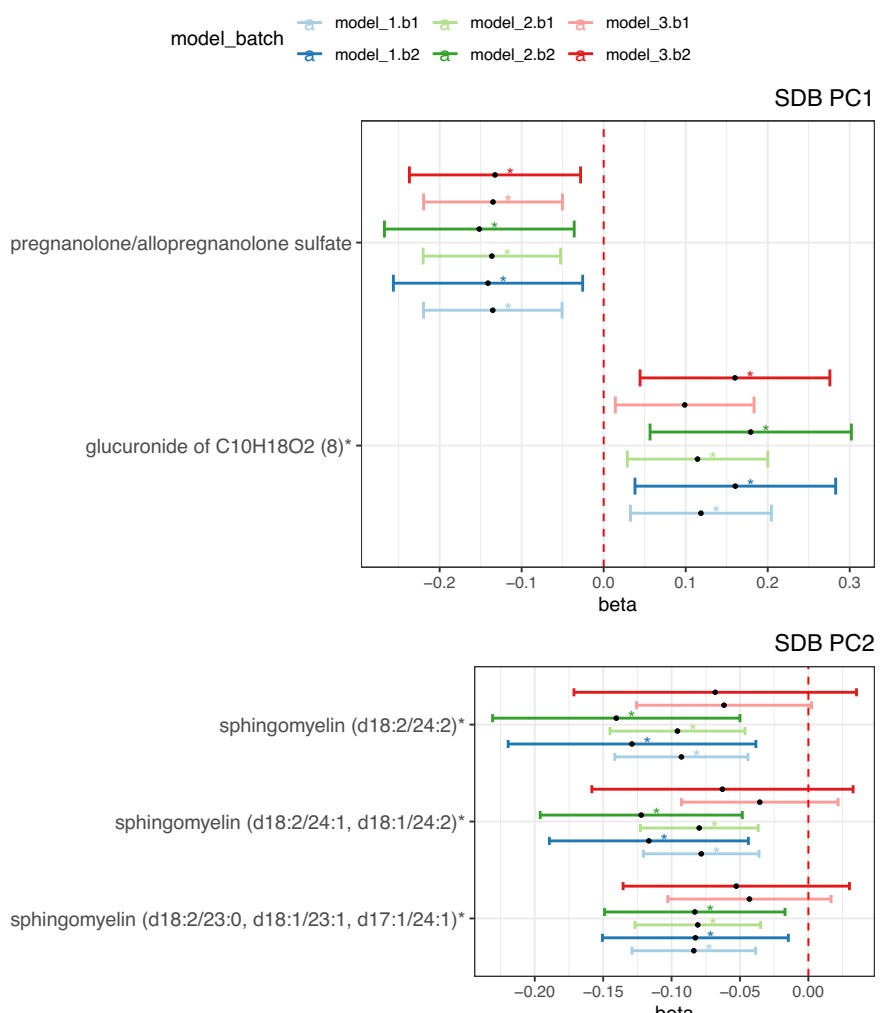

**Fig. 4 | Associations between SDB PCs and metabolites replicated in batch 2.**
SDB PC1: the first principal component of the seven sleep disordered breathing traits included in the principal component analysis; SDB PC2: the second principal component of the seven sleep disordered breathing traits included in the principal component analysis. Model 1 adjusted for demographic variables, including age, sex, field center, Hispanic/Latino background (Mexican, Puerto Rican, Cuban, Central American, Dominican, and South American and other/multi) and body mass index (BMI). Model 2 adjusted for all model 1 covariates and lifestyle variables – alcohol use, cigarette use, total physical activity (MET-min/day), and diet (Alternative Healthy Eating Index 2010) in addition to demographic variables. Model 3 adjusted for all model 1 and 2 covariates and comorbidities - indicators for diabetes and hypertension, and continuous measures of fasting insulin, fasting glucose, HOMA-IR, HDL, LDL, total cholesterol, triglycerides, systolic blood pressure and diastolic blood pressure. model_1.b1: model 1 in batch 1($n = 3299$ samples); model_2.b1: model 2 in batch 1($n = 3256$ samples); model_3.b1: model 3 in batch 1($n = 3182$ samples); model_1.b2: model 1 in batch 2($n = 1522$ samples); model_2.b2: model 2 in batch 2($n = 1500$ samples); model_3.b2: model 3 in batch 2 ($n = 1457$ samples). * indicates FDR-corrected $p < 0.05$ in batch 1 and FDR-corrected one-sided $p < 0.05$ in batch 2. Data are presented as effect estimates with 95% confidence intervals. P-values were derived from 1 degree-of-freedom Wald test, followed by False Discovery Rate Control (FDR). Batch 1 analysis used two sided tests and Batch 2 analysis used one sided tests. Exact $p$ values are provided in Supplementary Data 4. Metabolites with * indicate they were identified based on accurate mass data, retention time and mass spectrometry but not reference standards. Therefore, the verification is not as robust as metabolites without *. Source data are provided in Supplementary Data 18.

potentials for biomarkers[53,54]. Abnormalities in multiple lipid species have been implicated in sleep and circadian disruption[55]. Future studies are needed to understand the role of these metabolites in SDB.

The associations between these sphingomyelins and SDB PC2 were no longer statistically significant (FDR-corrected $p > 0.05$) in analysis adjusted for comorbidities (blood pressure-related phenotypes, diabetes and glycemic phenotypes, and cholesterol and lipid measures). Since diabetes mellitus and hypertension are common comorbidities to SDB, it is possible that these PC2-associated metabolites may partly reflect metabolic state related to these diseases, rather than being specific to SDB. Sphingolipids have been shown to mediate loss of insulin sensitivity, and to promote diabetic proinflammatory state, although the roles of specific sphingolipid species and pathways remain obscure[56].

It should be recognized also that the SDB PCs in this study were not constructed with the goal of developing new SDB phenotypes and did not utilize comprehensive polysomnography. Future work should build on emerging literature of subtypes of SDB[19,57,58] and characterize their metabolomics correlates and potential differences between them.

We found that metabolic scores reflecting SDB better predicted adverse cardiometabolic outcomes compared to the physiological phenotypes such as REI or those measuring hypoxia. We developed SDB MRSs, expanding our earlier work on OSA MRS[30]. We now further studied the association of the MRSs with incident cardiometabolic outcomes. Previous work in HCHS/SOL demonstrated that SDB was associated with incident hypertension and diabetes and insomnia was associated with incident hypertension[20]. Other studies also have

**Table 3 | Estimated associations between SDB PC metabolite risk scores and their respective phenotypes, in batch 1 and 2**

| | Both | | | Female | | | Male | | |
|---|---|---|---|---|---|---|---|---|---|
| | n | Coef [95%CI] | p | n | Coef [95%CI] | p | n | Coef [95%CI] | p |
| **Batch 1** | | | | | | | | | |
| SDB PC1 MRS | | | | | | | | | |
| Model 1 | 3299 | 0.29 [0.24, 0.34] | **2.03E-33** | 1874 | 0.31 [0.25, 0.36] | **1.55E-27** | 1425 | 0.26 [0.19, 0.33] | **1.44E-12** |
| Model 2 | 3256 | 0.30 [0.25, 0.36] | **4.75E-34** | 1854 | 0.31 [0.25, 0.36] | **2.72E-28** | 1402 | 0.26 [0.19, 0.33] | **9.72E-13** |
| SDB PC2 MRS | | | | | | | | | |
| Model 1 | 3299 | 0.23 [0.19, 0.28] | **6.04E-22** | 1421 | 0.24 [0.17, 0.32] | **7.71E-10** | 1425 | 0.23 [0.17, 0.28] | **1.12E-14** |
| Model 2 | 3256 | 0.23 [0.18, 0.28] | **5.50E-22** | 1406 | 0.23 [0.15, 0.30] | **1.20E-09** | 1402 | 0.22 [0.17, 0.28] | **2.35E-15** |
| Sex Specific SDB PC1 MRS | | | | | | | | | |
| Model 1 | | | | 1421 | 0.32 [0.26, 0.37] | **1.94E-32** | 1425 | 0.32 [0.26, 0.38] | **2.82E-26** |
| Model 2 | | | | 1406 | 0.32 [0.27, 0.37] | **5.74E-35** | 1402 | 0.32 [0.26, 0.37] | **8.50E-28** |
| Sex Specific SDB PC2 MRS | | | | | | | | | |
| Model 1 | | | | 1421 | 0.13 [0.06, 0.19] | **1.12E-04** | 1425 | 0.23 [0.17, 0.30] | **2.77E-12** |
| Model 2 | | | | 1406 | 0.10 [0.04, 0.17] | **1.70E-03** | 1402 | 0.22 [0.16, 0.29] | **4.38E-12** |
| **Batch 2** | | | | | | | | | |
| SDB PC1 MRS | | | | | | | | | |
| Model 1 | 1522 | 0.15 [0.08, 0.23] | **1.18E-04** | 960 | 0.09 [0.01, 0.19] | 6.26E-02 | 562 | 0.20 [0.10, 0.31] | **1.37E-04** |
| Model 2 | 1500 | 0.15 [0.07, 0.23] | **2.29E-04** | 950 | 0.08 [−0.01, 0.17] | 9.28E-02 | 552 | 0.20 [0.10, 0.30] | **5.50E-05** |
| SDB PC2 MRS | | | | | | | | | |
| Model 1 | 1484 | 0.14 [0.05, 0.22] | **1.54E-03** | 941 | 0.13 [0.01, 0.26] | **3.72E-02** | 562 | 0.14 [0.05, 0.23] | **2.22E-03** |
| Model 2 | 1463 | 0.14 [0.06, 0.22] | **4.57E-04** | 932 | 0.13 [0.02, 0.24] | **2.50E-02** | 552 | 0.15 [0.05, 0.25] | **2.32E-03** |
| Sex Specific SDB PC1 MRS | | | | | | | | | |
| Model 1 | | | | 941 | 0.08 [0.00, 0.15] | **4.53E-02** | 562 | 0.13 [0.04, 0.21] | **2.21E-03** |
| Model 2 | | | | 932 | 0.07 [−0.01, 0.14] | 7.52E-02 | 552 | 0.13 [0.04, 0.21] | **2.63E-03** |
| Sex Specific SDB PC2 MRS | | | | | | | | | |
| Model 1 | | | | 941 | 0.03 [−0.09, 0.14] | 6.61E-01 | 562 | 0.12 [0.02, 0.22] | **1.86E-02** |
| Model 2 | | | | 932 | 0.03 [−0.07, 0.13] | 4.92E-01 | 552 | 0.13 [0.04, 0.22] | **5.17E-03** |

Coef: coefficients are estimates of the incident rate ratio per 1 SD increase in a metabolite risk score (MRS). *p*-values were derived from a two-sided, 1 degree-of-freedom Wald test. Bolded values in the *p* column indicate their values are below 0.05. Model 1 adjusted for demographic variables, including age, sex, field center, Hispanic/Latino background and body mass index (BMI); Model 2 adjusted for all model 1 covariates and lifestyle variables – alcohol use, cigarette use, total physical activity (MET-min/day), and diet (Alternative Healthy Eating Index 2010) in addition to demographic variables.
SDB PC1 MRS/SDB PC2 MRS: metabolite risk scores developed based on coefficients from LASSO regression trained in both sex strata combined to predict the outcome (SDB PC1, or SDB PC2) in the discovery dataset (batch 1). Sex-specific SDB PC1 MRS/PC2 MRS: metabolite risk scores developed based on coefficients from LASSO regression trained in each sex stratum to predict the outcome (SDB PC1, or SDB PC2) in the discovery dataset (batch 1).

demonstrated associations between SDB with cardiometabolic and cardiovascular disorders[59,60]. Here, we saw that MRSs had stronger associations with incident diabetes and hypertension compared to measured physiological traits (REIs, hypoxia-related metrics, and SDB PCs), suggesting that the plasma-based SDB-related metabolites may be better markers of cardiometabolic risk than are physiological metrics made from a single overnight sleep study. While we are unable to verify this hypothesis using existing data, one explanation for this discrepancy is that there is variability in phenotypes generated from single-night polygraphy[61], reducing the predictive ability of derived physiological traits. In contrast, metabolomic profiles constructed as MRS may be more stable. In addition, MRS may better describe the metabolomic environment compared to SDB phenotypes that focus on breathing-related variables. Further, when examining only individuals with normal glycemic levels at baseline, the SDB PC2 MRS exhibited a more robust association with incident diabetes compared to the MRS derived using a simpler OSA phenotype (a binary measure SDB; null association in the analysis). This suggests a promising role for the SDB PC2 MRS for identifying individuals with SDB at elevated risks of developing diabetes before the onset of glucose dysregulation (i.e., early-stage diabetes). Given the null findings of many SDB intervention trials who recruited patients on the basis of physiological traits, future studies can evaluate the use of metabolic markers for identifying individuals who may benefit from SDB treatment.

Strengths of this study include the use of a large population of under-studied Hispanic/Latino adults, large panel of measured metabolites, rigorous analysis including a replication study, and assessment of association of constructed MRSs as well as of traditional SDB severity measures with incident hypertension and diabetes. The study also has a few limitations. Our study population is based on the HCHS/SOL cohort, which was designed to be representative of the diverse Hispanic/Latino population in the U.S. Historically, this population has been under-represented in research despite its risk for several adverse health outcomes. Future work should be done in other studies and populations to increase the generalizability of the findings. It is worth noting that in our previous study, OSA-metabolomics associations were replicated in a different racial minority cohort, Multi-Ethnic Study of Atherosclerosis (MESA)[30]. In this work we did not attempt to replicate the observed associations in MESA, because of the limited number of metabolites common in both HCHS/SOL and MESA. We used PCA, a linear dimension reduction method. Despite its advantage of interpretability, it could be less flexible than other non-linear techniques, and may not be an optimal method if the underlying structure among SDB phenotypes is non-linear. While it is attractive to interpret the SDB PCs as newly proposed clinical SDB measures, it is important to clarify that they were utilized to capture the variance of multiple SDB phenotypes in our dataset in a parsimonious manner rather than as

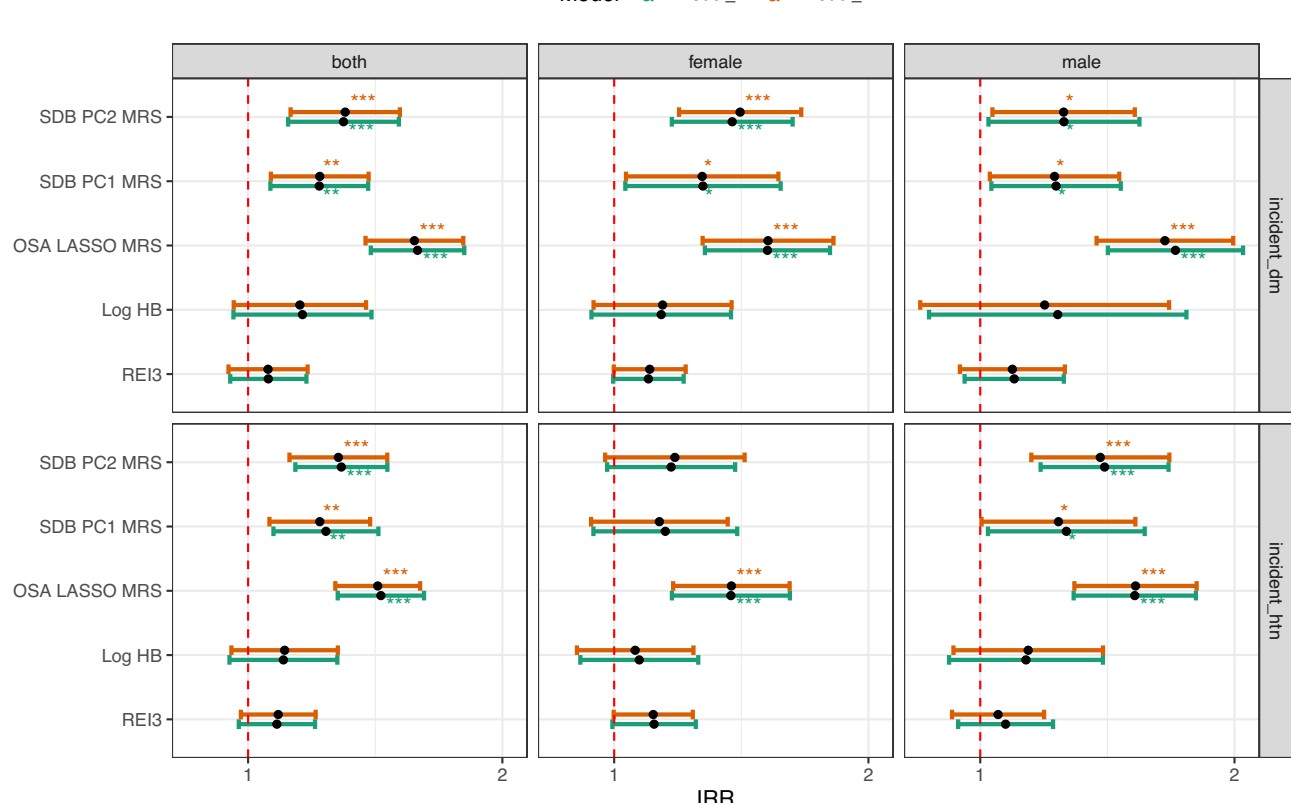

**Fig. 5 | Association between SDB phenotypes and incident cardiometabolic outcomes in the combined batch.** * indicates $p < 0.05$. ** indicates $p < 0.01$. *** indicates $p < 0.001$. Incidence rate ratios (IRR) were calculated using survey-weighted generalized linear regressions for each pair of cardiometabolic outcome (i.e., incident diabetes, incident hypertension) and SDB related phenotypes as the predictor in the combined dataset (batch 1 and batch 2), and are presented as effect estimates with 95% confidence intervals (CIs). $p$ values were derived from a one degree-of-freedom Wald test. Exact IRR, 95% CIs and $p$ values are provided in Supplementary Data 14. Model 1 adjusted for demographic variables, including age, sex, field center, Hispanic/Latino background (Mexican, Puerto Rican, Cuban, Central American, Dominican, and South American and other/multi) and body mass index (BMI). Model 2 adjusted for all model 1 covariates and lifestyle variables – alcohol use, cigarette use, total physical activity (MET-min/day), and diet (Alternative Healthy Eating Index 2010) in addition to demographic variables. REI3 Respiratory Event Index (REI) computed over all respiratory events, defined as apneas or hypopneas with at least 50% cannula flow reduction for a minimum duration of 10 s with >=3% oxygen desaturation; HB hypoxic burden; SDB PC1 MRS: metabolite risk score calculated based on the coefficients from LASSO regression trained in both sexes combined to predict SDB PC1 in the discovery dataset (batch 1); SDB PC2 MRS metabolite risk score calculated based on the coefficients from LASSO regression trained in both sexes combined to predict SDB PC2 in the discovery dataset (batch 1); OSA LASSO MRS: metabolite risk score calculated based on coefficients from LASSO regression trained to predict OSA in previous publication[30]; incident_dm: Incident diabetes (fasting glucose >=126 mg/dL, or post-OGTT glucose >=200 mg/dL or A1C > = 6.5%, or self-report of diabetes), $n = 2908$ samples for Model 1 and $n = 2874$ samples for Model 2; incident_htn: incident hypertension (systolic or diastolic blood pressure is greater than or equal to 140/90 or participant self-reported as currently taking antihypertensive medications), $n = 2388$ samples for Model 1 and $n = 2360$ samples for Model 2. Source data are provided in Supplementary Data 14.

new clinical measures. A new clinical measure should ideally be based on careful consideration of reliability, validity, and feasibility of each metric as made in clinical settings, where its predictive ability would be assessed. This is beyond the scope of our current study. Information loss may have occurred secondary to the use of rank normalization of the SDB phenotypes and the metabolite levels at a preprocessing step. The discovery and the replication datasets differed by age and several health characteristics, which may have reduced the ability to replicate findings. Given the observational nature of the study, and the cross-sectional association between the SDB PCs and metabolites, we cannot draw causal inferences. Thus, we cannot determine whether metabolite levels contribute to more severe SDB, or if SDB severity causes metabolomic changes, or if there are common mechanisms that influence both SDB and metabolites. Another limitation of this analysis is that some of the participants may have been treated for OSA. Less than 1% of the study participants (29 out of 3299 batch 1, and 17 out of 1522 batch 2 participants) reported "ever prescribed CPAP, BIPAP or oral device

treatment". Still, it is possible that some participants started treatment following the results from their home sleep apnea testing at the baseline examination. While it is unknown if these participants were actually treated or adhered to treatment, it is possible that OSA treatment in some study participants may have weakened the observed associations with incident cardiometabolic outcomes. Lastly, SDB MRSs do not include all the significantly associated metabolites, including the replicated metabolite pregnanolone/allopregnanolone sulfate, because only the metabolites in continuous format were used in the weighted sum forming the SDB MRSs.

Compared to our previous study[30] of OSA-metabolomics associations, the current analysis focuses on an expanded dataset, consistent of two batches from the HCHS/SOL, whereas the previous analysis used the first HCHS/SOL batch and a different study, MESA. In contrast to the previous analysis, both HCHS/SOL batches used the same metabolomics platform, allowing for investigation of a substantially larger number of metabolites. Other differences from the

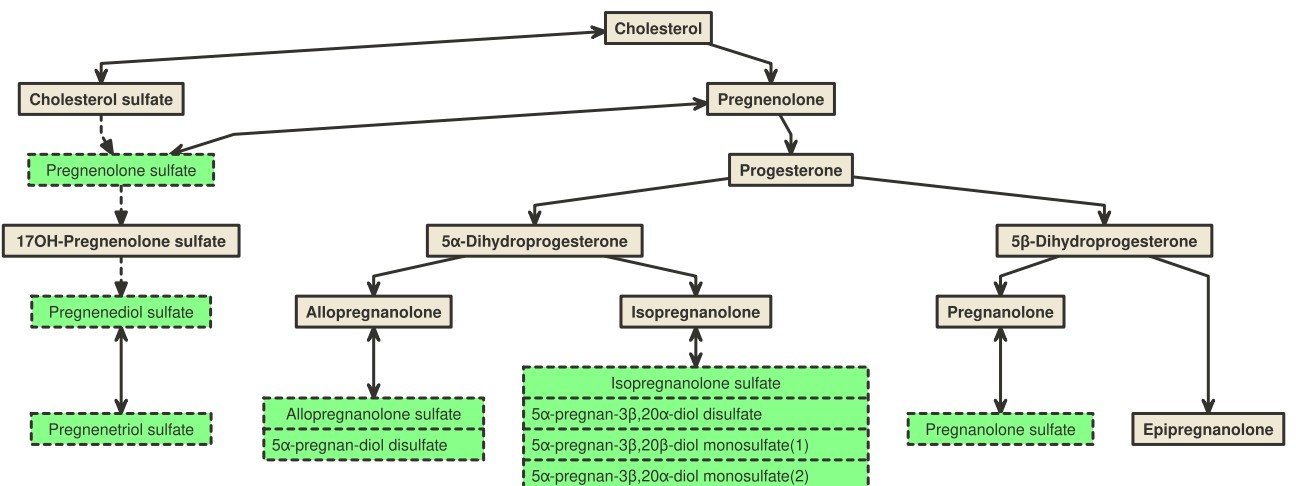

**Fig. 6 | Biosynthesis of progesterone steroids.** Metabolites highlighted in green indicate FDR-corrected *p* < 0.05 in the single metabolite association (SMA) analysis for SDB PCs in the discovery dataset (batch 1) in Model 1 adjusted for demographic variables, including age, sex, field center, Hispanic/Latino background (Mexican, Puerto Rican, Cuban, Central American, Dominican, and South American and other/multi) and body mass index (BMI).

previous publication include a different set of phenotypes, and the additional study of MRS associations with incident cardiometabolic outcomes. We report the associations of the 7 metabolites associated with OSA in HCHS/SOL in the previous manuscript (notably, of these only 4 replicated in MESA), with SDB phenotypes (Supplementary Data 9). All metabolites had evidence of association with at least one of the SDB PCs in at least one of the HCHS/SOL batches. However, only two associations passed FDR correction in batch 1 analysis in the current SDB analysis. It is important to consider the implications of these results for the development of clinical metabolomics biomarkers. Importantly, both the specific phenotypes and the set of metabolites used in association analyses are important and variation of either phenotypes or metabolites can lead to very different results. Similarly, the set of metabolites considered in the joint regression (LASSO or another approach) analysis may also result in identification of different biomarkers. While this may indicate a potential limitation of biomarker development using untargeted metabolomics, it also suggests opportunities to identify metabolite associations and develop biomarkers for subphenotypes of a disease, e.g., for SDB that is driven by specific endotypes. It also highlights the importance of replication analyses and for the accumulation of evidence across multiple studies, followed by rigorous targeted studies.

To summarize, using a discovery-replication study design, we identified and replicated multiple metabolites associated with SDB after being corrected for multiple comparisons. We constructed SDB MRSs which exhibited stronger associations with cardiometabolic sequalae of SDB, compared with physiologic SDB measures, including after accounting for demographic and lifestyle factors. These findings provide a strong basis for the use of metabolomics in studying SDB, including for clarifying and measuring risks for incident outcomes by different quantitative SDB phenotypes and dichotomous subtypes. Future evaluations are needed to study the use of MRSs for risk stratification, treatments response, and ultimately as biomarkers that guide diagnosis and treatment decisions.

## Methods
### Ethics statement
The HCHS/SOL was approved by the institutional review boards (IRBs) at each field center, where all participants gave written informed consent, and by the Non-Biomedical IRB at the University of North Carolina at Chapel Hill, to the HCHS/SOL Data Coordinating Center. All IRBs approving the HCHS/SOL study are: Non-Biomedical IRB at the

University of North Carolina at Chapel Hill. Chapel Hill, NC; Einstein IRB at the Albert Einstein College of Medicine of Yeshiva University. Bronx, NY; IRB at Office for the Protection of Research Subjects (OPRS), University of Illinois at Chicago. Chicago, IL; Human Subject Research Office, University of Miami. Miami, FL; Institutional Review Board of San Diego State University, San Diego, CA. All methods and analyses of HCHS/ SOL participants' materials and data were carried out in accordance with human subject research guidelines and regulations. This work was approved by the Mass General Brigham IRB and by the Beth Israel Deaconess Medical Center Committee on Clinical Investigations.

### The Hispanic Community Health Study/Study of Latinos
The Hispanic Community Health Study / Study of Latinos (HCHS/SOL) is a prospective community-based cohort study of 16,415 Hispanic/ Latino individuals aged 18–74 years at the baseline examination (2008-2011)[62]. Individuals were selected into the study using a multi-stage stratified random sampling from four geographic regions: Bronx NY, Chicago IL, Miami FL, and San Diego CA. The sampling strategy and study design were previously described[63]. Of study participants, 12,803 individuals were genotyped. Gender was self-reported according to a multiple-choice question with "male" and "female" as potential responses, and was later verified as biological sex for all genotyped individuals. Fasting blood samples were collected at the baseline examination. Within a week of the baseline examination in the clinic, 14,440 individuals were assessed for SDB using a validated Type 3 home sleep apnea test (ARES Unicorder 5.2; B-Alert, Carlsbad, CA) that measured nasal air-flow, position, snoring, heart rate and oxyhemoglobin saturation[3]. Among the baseline HCHS/SOL participants, 11,623 returned to a second clinic visit (visit 2) from 2014 to 2017, on average 6 years after the first visit.

### Metabolomics profiling and quality control
Of HCHS/SOL participants from the baseline examination who also had genetic data, 4004 individuals were selected at random for metabolomics profiling of fasting serum samples collected at baseline (metabolomics batch 1, processed in 2017). In 2021, additional 2368 serum samples from 2330 participants, also collected at baseline, were profiled in a second metabolomics batch 2. The second batch metabolomics data measured in HCHS/SOL was obtained based on three criteria. First, it included 50 samples for quality control analysis between the first and second batch. Second, it added individuals who

did not have metabolomics measured in the first batch. Including, it sampled a subset of individuals who participated in the ECHO-SOL ancillary study[64]. The ECHO-SOL ancillary study participants were a bit older on average compared to the entire cohort: a participant had to be at least 45 years old at the time of ECHO exam, where the ECHO-SOL exam took up to 3 years after a participant's baseline exam. It also sampled at random individuals with eGFR measures available from both the baseline and second HCHS/SOL exams.

Serum samples were stored at −70 °C at the HCHS/SOL Core Laboratory at the University of Minnesota until analysis by Metabolon, Inc. (Durham, NC) in 2017 (batch 1) and 2021 (batch 2). Serum samples were then extracted and prepared using Metabolon's standard solvent extraction method. Prior to extraction, samples were split into equal parts for untargeted analysis on both the gas chromatography-mass spectrometry and liquid chromatography-mass spectrometry (GC-MS and LC-MS)-based metabolomic quantification platforms[65,66]. Instrument variability was determined by calculating the median relative standard deviation (SD) for the internal standards added to each sample prior to injection into the mass spectrometers. Overall process variability was determined by calculating the median relative SD for all endogenous metabolites (i.e., non-instrument standards) present in 100% of the technical replicate samples. Metabolite quantitation was relative, so that relationship between metabolite levels between metabolites and between individuals in the same sample are consistent, but the specific metabolite concentration for a given person and a given metabolite values may be different from those obtained by an absolute quantitation method. Detailed methods of metabolite assaying by Metabolon, Inc, are provided in Supplementary Note 1.

We took a discovery-and-replication approach using batch 1 as the discovery and batch 2 as the replication dataset. Preprocessing of the metabolomic data is described in Supplementary Fig. S6. First, we removed batch 2 individuals who overlapped with batch 1 and duplicated samples from the same individuals, resulting in 2178 remaining observations. Next, we kept metabolites that were known and available in both batches and excluded xenobiotics. We also excluded metabolites with missing values in more than 75% of the individuals in either batch 1 or batch 2. Metabolites with missing values in 25 – 75% of the individuals in both batches were dichotomized as "observed" and "not observed" – referred to as "dichotomized metabolites" henceforth). Metabolites that had different missingness patterns between the batches (e.g., < 25% missing values in one batch and > 25% missing value in the second batch) were excluded. For metabolites with missing values in up to 25% of the individuals in both batches, we assumed that missing values were due to concentrations below the minimum detection limits, thus imputed the missing values for each metabolite with the lowest non-missing value of that metabolite across the samples within the batch. We then rank-normalized these metabolite measures in each batch separately. In the sex-stratified analysis, we used the same rank-normalized metabolites (and did not rank-normalize within sex groups).

### Sleep disordered breathing phenotypes

We selected seven available, correlated SDB phenotypes, previously studied by our group and others in the context of SDB genetics and omics[20,23,33], and were shown to capture potentially different characteristics of SDB and its association with clinical outcomes. These included Respiratory Event Index 0 (REI0), the sum of all respiratory events (apneas or hypopneas with at least 50% airflow flow reduction for a minimum duration of 10 s), regardless of oxygen desaturation, divided by estimated sleep time; the sum of all respiratory events associated with >=3% oxygen desaturation divided by estimated sleep time (REI3); respiratory event duration (the average length of each respiratory event); sleep-apnea associated hypoxic burden[21]; the minimum and the average oxyhemoglobin saturation

during the sleep period; and the percentage of estimated sleep period with oxyhemoglobin saturation below 90%. We then performed sampling-weighted Principal Component Analysis (PCA), accounting for the HCHS/SOL study design, over the complete HCHS/SOL study population with non-missing SDB measures. The goal of the PCA was to capture the most substantial variance among the multiple, correlated, SDB phenotypes in our dataset while reducing the dimensionality of the data, allowing for more parsimonious modeling. We rank-normalized the 7 SDB measures prior to PCA analysis due to the highly non-normal distribution of some of the measures and so that measures with wide range do not dominate the PCA results. We used the PCs that explain at least 10% of the variance in the SDB measures in the subsequent analyses. As a sensitivity analysis, we also implemented PCA among the batch 1 participants, and derived SDB PCs for the batch 2 participants based on the loadings from batch 1, with SDB phenotypes rank-normalized within each batch separately.

To interpret SDB phenotypes captured by the PCs, we characterized the study populations defined by the low and high 10% values of each of the PCs selected for further analysis. Characteristics include demographic (age, sex), cardiometabolic (BMI, hypertension, diabetes), and sleep measures (SDB and self-reported insomnia, sleep duration, sleepiness) variables.

### Model covariates

All analyses used up to three conceptual models. Model 1 (i.e., base model) adjusted for demographic variables, including age, sex, field center, Hispanic/Latino background (Mexican, Puerto Rican, Cuban, Central American, Dominican, and South American and other/multi), and body mass index (BMI). Hispanic/Latino background was included because cultural differences between groups are potentially associated with differences in diet, which is highly associated with levels of many metabolites. Model 2 is further adjusted for lifestyle variables – alcohol use, cigarette use, total physical activity (MET-min/day), and diet (Alternative Healthy Eating Index 2010) in addition to demographic variables. Model 3 is a lifestyle and comorbidity model that is adjusted for Model 2 variables, and in addition, for continuous measures of fasting insulin, fasting glucose, HOMA-IR, HDL, LDL, total cholesterol, triglycerides, systolic blood pressure, diastolic blood pressure, as well as for indicators of diabetes and hypertension, which encapsulate additional information about medication use, not captured by continuous measures.

### Single metabolite associations (SMA) between individual metabolites and SDB PCs

Using survey-weighted generalized linear regressions, each metabolite's concentration level was regressed separately against SDB PC outcomes, with a recognition that cross-sectional data cannot establish a causal direction. We used the Benjamini-Hochberg method[67] to control false discovery rate (FDR) for multiple testing among metabolites in all models for each SDB PC in batch 1. Metabolites were flagged for further validation if the FDR-corrected $p$ < 0.05 in Model 1, for either SDB PC1 or PC2. In the replication analysis, we tested the associations of these flagged metabolites with SDB PCs in linear regression models in batch 2 in Models 1–3. We computed one-sided $p$-values guided by the estimated directions of the associations in batch 1[68], and determined replication if the FDR-corrected one-sided $p$-value was <0.05. To further understand the SMA results in relations to the original SDB phenotypes, we also conducted the SMA between individual metabolites with replicated associations with the SDB PCs and the 7 individual SDB phenotypes (rank-normalized) individually.

In a follow-up analysis, we visualized the concentrations of raw and rank-normalized metabolites from sex hormone-related pathways that were associated with SDB by sex and age strata.

To evaluate whether metabolites reported as associated with OSA in our previous study in HCHS/SOL and MESA[30], we extracted and described their estimated associations with the SDB PCs in both batches.

### LASSO regression for constructing the SDB metabolite risk scores (SDB MRS)

For each SDB PC, we applied LASSO linear regression over all 582 continuously modeled metabolites, adjusted for the covariates from Model 1 (unpenalized) in HCHS/SOL batch 1. We selected the LASSO tuning parameter by minimizing the prediction error for SDB PCs in a 10-fold cross-validation. SDB MRSs were calculated as weighted sums of the normalized metabolite serum concentrations, with weights being the metabolite coefficients from the LASSO regression from batch 1. In association analyses using the MRSs, we standardized (z-scored) them to have mean of 0 and variance of 1 using the sample mean and variance (Supplementary Data 9).

To validate the associations between the SDB MRS with SDB PCs, we constructed the SDB PC1 MRS and SDB PC2 MRS in batch 2 using the weights from the LASSO regression conducted in batch 1, then assessed their associations with the corresponding SDB PCs in Models 1–3. In secondary analyses we assessed potential sex differences via: (1) sex-stratified SDB MRSs constructed based on sex-stratified LASSO; (2) sex-stratified association analyses for sex-specific and sex-combined SDB MRSs. We also assessed the associations between SDB MRS quartiles and the corresponding SDB PCs.

As a sensitivity analysis, we assessed the robustness of the developed MRSs by limiting potential impacts of sample overlap and of medication use. Thus, we applied LASSO regression to develop MRSs using a new set of SDB PCs developed exclusively among the batch 1 participants who were not taking any antihypertensives or antidiabetics at the baseline exam.

### Incident outcomes

We also studied the associations of the SDB PCs and their MRSs with incident hypertension and diabetes, assessed at visit 2, among individuals free of hypertension and free of diabetes, respectively, at the baseline exam. Diabetes was determined based on American Diabetes Association (ADA) definition or scanned medication (at the baseline exam) or self-reported diabetes medication use (at the second exam). ADA criteria are based on laboratory tests – fasting glucose >=126 mg/dL, or post-OGTT glucose >=200 mg/dL or A1C > = 6.5%[69]. In a secondary analysis, incident diabetes was assessed separately among individuals with impaired glucose tolerance (fasting glucose within 100–125 mg/dL, or post-OGTT glucose within 140–199 mg/dL, or A1C within 5.7–6.5%) and among normal glycemic individuals. Hypertension was determined following the NHANES guidelines: systolic or diastolic blood pressure is greater than or equal to 140/90 or participant self-reported as currently taking antihypertensive medications[70].

### Association analyses between SDB phenotypes and incident cardiometabolic outcomes

Finally, survey-weighted Poisson regressions were implemented to assess the associations between incident hypertension and diabetes among batch 1 and 2 combined study samples with various SDB phenotypes including benchmark singular sleep measures (i.e., REI 3%, hypoxic burden) and our newly developed composite measures (i.e., SDB PCs, and SDB MRSs), as well as our recently developed OSA MRS, adjusting for Model 1 and 2 covariates, respectively. The OSA MRS was trained using LASSO on moderate to severe OSA (defined as REI3 > = 15) in the HCHS/SOL cohort and previously validated in the MESA cohort[30]. We combined the two batches in this analysis to increase statistical power by having a larger sample size. To combine metabolomics data of batch 1 and batch 2 we aggregated the metabolites from non-overlapping batch 1 and batch 2 individuals, after imputation and rank-normalization of each metabolite separately in each batch.

In a sensitivity analysis, we estimated the associations between the SDB PCs MRS developed in a process (both PCA and LASSO) involving only batch 1 participants who were not taking any antihypertensives or antidiabetics at baseline and incident cardiometabolic outcomes within batch 2 participants who also did not take antihypertensives or antidiabetics at baseline.

All analyses were done in R 3.6.3. svyglm was used for survey-weighted generalized linear regression models, and svyprcomp was used for sampling-weighted principal component analysis, both of which were from the survey package (version 4.1)[71]. The glmnet R package (version 3.0)[72] was used for the LASSO linear regression.

### Reporting summary

Further information on research design is available in the Nature Portfolio Reporting Summary linked to this article.

## Data availability

HCHS/SOL data are available through application to the data base of genotypes and phenotypes (dbGaP) accession phs000810. HCHS/SOL metabolomics data are available via data use agreement with the HCHS/SOL Data Coordinating Center (DCC) at the University of North Carolina at Chapel Hill, see collaborators website: https://sites.cscc.unc.edu/hchs/. Researchers can email the HCHS/SOL DCC at hchsadministration@unc.edu. The metabolite association data generated in this study are provided in the Supplementary Information. Source data for results presented in the figures are provided with this paper.

## Code availability

The code used for conducting the analyses, as well as for the generation of figures and tables presented in this study, are written in R, and available at https://github.com/yzhang104/HCHS-SOL_SDB_metabolomics.git[73].

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

## Acknowledgements

The authors thank the staff and participants of HCHS/SOL for their important contributions. Investigators website - http://www.cscc.unc.edu/hchs/. This work was supported by National Heart Lung and Blood Institute grants R01HL161012 to T.S., R35HL135818 to S.R., and K24HL13632 to P.L. Support for metabolomics data was graciously provided by the JLH Foundation (Houston, Texas). The Hispanic Community Health Study/Study of Latinos is a collaborative study supported by contracts from the National Heart, Lung, and Blood Institute (NHLBI) to the University of North Carolina (HHSN268201300001I/N01-HC-65233), University of Miami (HHSN268201300004I/N01-HC-65234), Albert Einstein College of Medicine (HHSN268201300002I/N01-HC-65235), University of Illinois at Chicago (HHSN268201300003I/N01- HC-65236 Northwestern Univ), and San Diego State University (HHSN268201300005I/N01-HC-65237). The following Institutes/Centers/Offices have contributed to the HCHS/SOL through a transfer of funds to the NHLBI: National Institute on Minority Health and Health Disparities, National Institute on Deafness and Other Communication Disorders, National Institute of Dental and Craniofacial Research, National Institute of Diabetes and Digestive and Kidney Diseases, National Institute of Neurological Disorders and Stroke, NIH Institution-Office of Dietary Supplements.

## Author contributions

Conception: T.S. and Y.Z.; HCHS/SOL metabolomics data collection and design: B.Y., Q.Q. and E.B.; Sleep study design: S.R. and P.C.Z.; Data analysis: Y.Z. and T.S.; HCHS/SOL data collection and management: P.C.Z., J.C., M.L.D. and R.K.; Results interpretation and critical review of manuscript: Y.Z., B.Y., Q.Q., A.A., H.C., N.A.S., A.R.R., P.C.Z., J.C., M.L.D., E.B., R.K., P.Y.L., S.R. and T.S.

## Competing interests

S.R. has received consulting fees from Eli Lilly Inc and is an advisor to ApniMed Inc. This work is not related to this paper. A.A. reports grant support from Somnifix and serves as a consultant for Somnifix, Respicardia, Eli Lilly and Apnimed. Apnimed is developing pharmacological treatments for Obstructive Sleep Apnea. A.A. reports patent application related to the measurement of hypoxic burden of sleep apnea. A.A.'s interests were reviewed by Brigham and Women's Hospital and Mass General Brigham in accordance with their institutional policies. A.R.R. is a member of sleep disorders research advisory board for the NIH/NHLBI. P.C.Z. is past president of the World Sleep Society, consultant for Eisai, Idorsia, CVS Caremark and Jazz. This work is not related to this paper. Y.Z., B.Y., Q.Q., H.C., N.A.S., J.C., M.L.D., E.B., R.K., P.Y.L. and T.S. have no competing interests as defined by Nature Research, or other interests that might be perceived to influence the interpretation of the article.

## Additional information

[1]Division of Sleep Medicine and Circadian Disorders, Department of Medicine, Brigham and Women's Hospital, Boston, MA 02115, USA. [2]Department of Epidemiology, School of Public Health, The University of Texas Health Science Center at Houston, Houston, TX 77030, USA. [3]Department of Epidemiology and Population Health, Albert Einstein College of Medicine, Bronx, New York, NY, USA. [4]Division of Sleep and Circadian Disorders, Departments of Medicine and Neurology, Brigham & Women's Hospital & Harvard Medical School, Boston, MA 02115, USA. [5]Human Genetics Center, Department of Epidemiology, School of Public Health, The University of Texas Health Science Center at Houston, Houston, TX 77030, USA. [6]Department of Medicine, Icahn School of Medicine at Mount Sinai, New York, NY 10029, USA. [7]Sleep Medicine Program, Department of Neurology, University of Miami Miller School of Medicine, Miami, FL 33136, USA. [8]Division of Sleep Medicine, Department of Neurology, Northwestern University, Chicago, IL 60611, USA. [9]Collaborative Studies Coordinating Center, Department of Biostatistics, University of North Carolina at Chapel Hill, Chapel Hill, NC 27599, USA. [10]Department of Preventive Medicine, Northwestern University Feinberg School of Medicine, Chicago, IL 60612, USA. [11]Public Health Sciences Division, Fred Hutchinson Cancer Research Center, Seattle, WA 98109, USA. [12]The Institute for Translational Genomics and Population Sciences, The Lundquist Institute at Harbor-UCLA Medical Center, Torrance, CA 90502, USA. [13]Department of Biostatistics, Harvard T.H. Chan School of Public Health, Boston, MA 02115, USA. [14]CardioVascular Institute, Beth Israel Deaconess Medical Center, Boston, MA 02115, USA. ✉e-mail: tsofer@bidmc.harvard.edu

