## [Peer Review File · Nature Communications]

Metabolomic Profiles of Sleep-Disordered Breathing are Associated with Hypertension and Diabetes Mellitus DevelopmentREVIEWER COMMENTS

Reviewer #1 (Remarks to the Author):

This manuscript assesses the associations between sleep-disordered breathing phenotypes and metabolomic profiles. Strengths include a large prospective community-based cohort with metabolomic and sleep data as well as longitudinal outcomes, as well as analysis with a replication cohort. I commend the authors on their analysis.

1. My primary concern is the validity of the new SDB measures PC1 and PC2. The authors construct these using principal component analysis of 7 different SDB measures. This method is appropriate to capture the variability within the dataset, but they do not appear to be measures that have been previously studied or validated outside of the dataset. There is unknown clinical significance of these measures, especially since PC2 was not significantly associated with either incident DM or HTN.
2. It was also unclear the rationale for the 7 SDB measures analyzed, especially REI3 and REI0 versus more commonly used measures of apnea and hypopnea.
3. The definition for identifying SDB (versus OSA, which was defined as REI >15) in the HCHS/SOL cohort is not clearly stated. A CONSORT diagram of the HCHS cohort for the analysis would be helpful.
4. Was there any hypotheses at the start of the study for what metabolomics might be implicated in SDB or in the sex-stratified analysis?
5. While the HCHS/SOL is an important cohort to study given the understudied population, further discussion could be made about whether there is external generalizability outside of this cohort for the metabolomic associations.
6. Figure 2 legibility would be improved by using clearer variable labels rather than the variable name used for data analysis.

Reviewer #2 (Remarks to the Author):

This manuscript reports findings from The Hispanic Community Health Study/Study of Latinos (HCHS/SOL) focused on identifying potential metabolomics based biomarkers that represent a composite of several measures related to sleep disordered breathing (SDB).

From a metabolomics perspective a strength of the work is the use of independent training and replication datasets which helps add confidence to the associations reported between metabolites and SDB. This notion of identifying metabolites that may represent a composite of SDB measures is novel and has potential to help advance the field. However, there are notable limitations to the work that minimize the clinical implication of the identified biomarkers. Notably, the cross-sectional study design precludes inferring direction of causality in the observed associations. Please see below comments for key considerations.

MAJOR:

1) The authors note in their prior work (citation #30) "we identified metabolites associated with moderate to severe OSA (defined as a Respiratory Event Index [REI] \geq 15) and constructed an index composed of 14 metabolites, associated with OSA cross-sectionally, in two independent datasets.". It is not clear if any of the metabolites from this prior work in citation #30 were replicated in the current analyses. If there were any overlapping metabolites can you please expand on this in the Discussion, and if not, are there any reasons why this work in different studies is not identifying similar metabolites? What are the implications of this, either way, for developing clinical biomarkers?

2) The metabolomics workflow is lacking some information. What was the process used for identifying/annotating metabolite names? Were there any unknown metabolites? Can you please provide the Metabolomics Standard Initiative numbers for level of confidence in the metabolite annotations? Please also provide a standard metabolite identification number such as HMDB, PubChem, Lipid Maps, etc. This information will be helpful for future replication studies. For example you identify a sphingomyelin as sphingomyelin(d18:2/23:0,d18:1/23:1,d17:1/24:1)...does this indicate a sphingomyelin with 3 potential different combinations of fatty acids?

3) Please clarify if any of the participants were undergoing treatment for SDB at time of sample collection or if such treatment could have influenced the analyses focused on incidence of diabetes and hypertension?

4) Related, did you collect information on medications participants were taking, especially

lipid lowering, anti-hypertensive, and glucose lowering medications? Could these potential medications influence the identified metabolites and incidence of diabetes and hypertension?

5) Please consider commenting on the fact that the PCA analysis of the SDB outcomes were conducted on all participants together, including the training and replication datasets. This means the PCS analyses already identified participants in both the training and replication datasets that have similar (or group together) clinical SDB features which may bias the metabolomics analyses to have a higher probability of replicating associations with the PC1 and PC2. If the PCA is conducted on the training set first, does it replicate in your replication dataset? This is an important consideration to determine if the PCA analysis, which all subsequent analyses are based on, is generalizable outside your study participants.

6) PC1 explains ~65% of variance whereas PC2 explains ~14% of the variance, does this imply the results and metabolites linked to PC1 have a higher probability of being more strongly related to SDB?

7) Model 1 includes body weight and BMI as covariates, do these variables represent independent characteristics or are they correlated with each other?

8) As stated on page 10 lines 214-219 "Model 3 is a lifestyle and comorbidity model that is adjusted for Model 2 variables in addition to indicators for diabetes mellitus and hypertension, and continuous measures of fasting insulin, fasting glucose, HOMA IR, HDL, LDL, total cholesterol, triglycerides, systolic blood pressure and diastolic blood pressure.". The continuous measures appear to be indicators for diabetes mellitus and hypertension, but as stated it seems there are other indicators for diabetes mellitus and hypertension in the model? Please clarify precisely what variables are in model 3.

9) Related, model 3 is minimally used in the analyses but could be highly informative. For example, do the variable metabolite biomarkers still associate with incidence of diabetes or hypertension when model 3 is used? This is important to understand if adding the metabolite biomarkers to traditional diabetes and hypertension clinical risk factors improves

model performance or risk prediction or if the traditional clinical risk factors already account for the variability/risk explained by the new biomarker models? This is especially important because your analyses with incidence of diabetes and hypertension used the training and replication datasets together and there was no validation assessment for these analyses, decreasing the rigor of the findings.

10) In the replication analyses with one-sided p-values it appears no FDR correction was applied. Please confirm and justify why no FDR correction was needed/used for these analyses, do the findings change if a FDR correction is used?

11) Please clarify if metabolite concentrations in your models were relative concentrations or absolute concentrations and what implication this has for translating biomarkers to clinical applications.

12) SDB PC2 MRS was associated with incident diabetes but SDB PC2 itself was not. This is intriguing as SDB PC2 MRS was designed to represent or capture SDB PC2. These divergent associations with incident diabetes are somewhat hard to interpret then, especially without a replication/validation set in these analyses. Could this be potentially related to PC2 only explaining a relatively small proportion of the SDB measures (~14%)? Does these divergent findings limit the potential clinical utility of these biomarkers?

13) Supplemental Table S12 is hard to read/interpret as uploaded for the reviewers, a minor comment.

14) Given the cross-sectional study design and lack of replication for the analyses with incident diabetes and hypertension, please more explicitly state that the direction of causality cannot be inferred in current analyses. It is plausible that more severe SDB could be driving the changes in metabolites as opposed to changes in metabolites driving more severe SDB. This is an important consideration for your statement in Discussion on page 29 lines 500-503: "If true, these data point to the possibility that different classes of steroids of different origins may be involved in the development of SDB, and its association with incident hypertension and diabetes mellitus."

Response to review of NCOMMS-23-30340 “Metabolomic Profiles of Sleep-Disordered Breathing are Associated with Hypertension and Diabetes Mellitus Development: the HCHS/SOL”

REVIEWER COMMENTS

Reviewer #1 (Remarks to the Author):

This manuscript assesses the associations between sleep-disordered breathing phenotypes and metabolomic profiles. Strengths include a large prospective community-based cohort with metabolomic and sleep data as well as longitudinal outcomes, as well as analysis with a replication cohort. I commend the authors on their analysis.

Response: Thank you!

1. My primary concern is the validity of the new SDB measures PC1 and PC2. The authors construct these using principal component analysis of 7 different SDB measures. This method is appropriate to capture the variability within the dataset, but they do not appear to be measures that have been previously studied or validated outside of the dataset. There is unknown clinical significance of these measures, especially since PC2 was not significantly associated with either incident DM or HTN.

Response:

We appreciate your thoughtful comments. Your feedback provides an excellent opportunity for us to clarify the intention behind our principal component analysis approach. We'd like to emphasize that our intent was not to propose SDB PC1 and PC2 as novel clinical measures of SDB. Rather, our usage of principal component analysis served the primary purpose of capturing the most variance associated with multiple SDB phenotypes in a parsimonious manner. Your comment on the clinical relevance of PC2, given its non-significant association with incident DM or HTN, is noted. It's worth emphasizing that the lack of association of a PC with certain outcomes like DM or HTN does not inherently invalidate its utility for the analytical purpose it served within our study. Metabolomic changes might be more sensitive than physiological changes captured in the SDB phenotypes, which may contribute to the more significant associations of the MRSs with incident DM and HTN despite the lack of statistical significance in the associations between SDB PCs and incident outcomes.

We also wanted to clarify that most of the SDB metrics we studied are used commonly in the clinically assessment of SDB (i.e., event frequency and measures of hypoxia) or have been shown in emerging research to be predictive of mortality. Note that event duration has been shown to predict mortality (PMID: 30336691) and is a marker of a SDB mechanistic trait of low

arousal threshold (PMID: 33285084). We included both REI0 (in addition to the more standard REI3 measure) given ongoing interest in how to best define respiratory events (e.g., it has been argued that women and individuals with pigmented skin may not exhibit the same degree of desaturation as men or white individuals).

Given your feedback, we recognize the need to ensure the readers understand the purpose and limitation of the PCA approach therefore we also made changes to method and discussion session.

Method section, page 9, line 17:

“The goal of the PCA was to capture the most substantial variance among the multiple, correlated, SDB phenotypes in our dataset while reducing the dimensionality of the data, allowing for more parsimonious modeling.”

Discussion section, p36, line 5:

“While it is attractive to interpret the SDB PCs as newly proposed clinical SDB measures, it is important to clarify that they were utilized to capture the variance of multiple SDB phenotypes in our dataset in a parsimonious manner rather than as new clinical measures. A new clinical measure should ideally be based on careful consideration of reliability, validity, and feasibility of each metric as made in clinical settings, where its predictive ability would be assessed. This is beyond the scope of our current study.”

2. It was also unclear the rationale for the 7 SDB measures analyzed, especially REI3 and REI0 versus more commonly used measures of apnea and hypopnea.

Response: we agree that these SDB measures are not ideal if our goal was to develop a new clinical SDB measure. SDB measurements were obtained using a validated home sleep apnea test (a modality currently used to diagnose more than 70% of patients in clinical settings.) Home sleep apnea tests are limited by the absence of EEG-based sleep assessment, precluding assessment of the apnea hypopnea index (AHI), the gold standard for OSA diagnosis. The term “REI” is the AASM-recommended term to capture an index that summarize apneas and hypopneas per hour of estimated sleep duration (and thus is the analogue to the AHI, does not include EEG-based sleep assessment). The various event-based oxygen saturation measures have been used in the past to study SDB by our group and others. Event duration is a relatively novel phenotype that our previous work (PMID: 34144485, 33285084) highlighted as important, as mentioned above. To address this comment, we now edited the sentence introducing these phenotypes.

Methods section, page 9 line 5:

“We selected seven available, correlated SDB phenotypes, previously studied by our group and others in the context of SDB genetics and omics, and were shown to capture

potentially different characteristics of SDB and its association with clinical outcomes.”

3. The definition for identifying SDB (versus OSA, which was defined as REI >15) in the HCHS/SOL cohort is not clearly stated. A CONSORT diagram of the HCHS cohort for the analysis would be helpful.

Response: In this analysis, we did not define SDB as a dichotomized measure, but rather we just used continuous measures of SDB without assigning individuals to groups. A CONSORT diagram is available as Supplementary Figure S1.

4. Was there any hypotheses at the start of the study for what metabolomics might be implicated in SDB or in the sex-stratified analysis?

Response: This analysis was discovery-driven. We could certainly have developed hypotheses based on numerous publications about OSA/SDB and metabolomics, but explicitly followed much of the literature that has utilized discovery-driven analyses and showed this approach to be very successful in identifying replicated genetic and omics associations with phenotypes. Such approaches posit that there is often new biologic insights that can be derived by a data-drive approach, as long as methods are in place to account for the multiple testing.

5. While the HCHS/SOL is an important cohort to study given the understudied population, further discussion could be made about whether there is external generalizability outside of this cohort for the metabolomic associations.

Response: Completely agree. We addressed it by adding the following sentence to the discussion section, page 35 line 16:

“Our study population is based on the HCHS/SOL cohort, which was designed to be representative of the diverse Hispanic/Latino population in the U.S. Historically, this population has been under-represented in research despite its risk for several adverse health outcomes. Future work should be done in other studies and populations to increase the generalizability of the findings. It is worth noting that in our previous study, OSA-metabolomics associations were replicated in a different racial minority cohort (MESA).”

6. Figure 2 legibility would be improved by using clearer variable labels rather than the variable name used for data analysis.

Response: Thank you for the comments, we completely agree. We have revised the variable labels in Figure 2.

Reviewer #2 (Remarks to the Author):

This manuscript reports findings from The Hispanic Community Health Study/Study of Latinos (HCHS/SOL) focused on identifying potential metabolomics based biomarkers that represent a composite of several measures related to sleep disordered breathing (SDB). From a metabolomics perspective a strength of the work is the use of independent training and replication datasets which helps add confidence to the associations reported between metabolites and SDB. This notion of identifying metabolites that may represent a composite of SDB measures is novel and has potential to help advance the field. However, there are notable limitations to the work that minimize the clinical implication of the identified biomarkers. Notably, the cross-sectional study design precludes inferring direction of causality in the observed associations. Please see below comments for key considerations.

Response: Thank you for your review and helpful comments.

MAJOR:

1) The authors note in their prior work (citation #30) "we identified metabolites associated with moderate to severe OSA (defined as a Respiratory Event Index [REI] \geq 15) and constructed an index composed of 14 metabolites, associated with OSA cross-sectionally, in two independent datasets.". It is not clear if any of the metabolites from this prior work in citation #30 were replicated in the current analyses. If there were any overlapping metabolites can you please expand on this in the Discussion, and if not, are there any reasons why this work in different studies is not identifying similar metabolites? What are the implications of this, either way, for developing clinical biomarkers?

Response: Thank you for the thoughtful comment. First, we would like to clarify that the previously published OSA analysis was different, in that it studied OSA association with a smaller set of ~210 metabolites that were in common to HCHS/SOL (a sample that is referred to as "batch 1" in the current manuscript) and the MESA study. The current analysis used a larger set of metabolites, but required that all metabolites were available in two separate batches of the HCHS/SOL. We clarified and discussed these differences in multiple places in the manuscript. Briefly, the results are provided in Supplementary Table S7, and all 7 OSA-reported metabolites have some evidence of association but only two had association SDB PC p-values passing FDR adjustment.

In the methods section, page 12 line 1:

"To evaluate whether metabolites reported as associated with OSA in our previous study in HCHS/SOL and MESA, we extracted and described their estimated associations with the SDB PCs in both batches."

In the results section, page 24, line 4:

"To compare the SDB SMA results with our prior publication reporting OSA-metabolite associations, we examined the overlap between the OSA and SDB PCs metabolite associations. While none of the metabolites associated with SDB PCs were included in the OSA SMA analysis, all metabolites reported as associated with OSA were included in

the present SDB SMA analysis (**Supplementary Table S7**). Briefly, all metabolites had some evidence of association with SDB PC1 (p-value < 0.1 in either batch), but only two metabolites had FDR-corrected association p-value < 0.05 in the batch 1 discovery analysis.

In a discussion paragraph, as suggested by the reviewer (page 37 line 7):

“Compared to our previous study of OSA-metabolomics associations, the current analysis focuses on an expanded dataset, consistent of two batches from the HCHS/SOL, whereas the previous analysis used the first HCHS/SOL batch and a different study, the Multi-Ethnic Study of Atherosclerosis (MESA). In contrast to the previous analysis, both HCHS/SOL batches used the same metabolomics platform, allowing for investigation of a substantially larger number of metabolites. Other differences from the previous publication include a different set of phenotypes, and the additional study of MRS associations with incident cardiometabolic outcomes. We report the associations of the 7 metabolites associated with OSA in HCHS/SOL in the previous manuscript (notably, of these 7, only 4 replicated in MESA), with SDB phenotypes (Supplementary Table S7). All metabolites had evidence of association with at least one of the SDB PCs in at least one of the HCHS/SOL batches. However, only two associations passed FDR correction in batch 1 analysis in the current SDB analysis. It is important to consider the implications of these results for the development of clinical metabolomics biomarkers. Importantly, both the specific phenotypes and the set of metabolites used in association analyses are important and variation of either phenotypes or metabolites can lead to very different results. Similarly, the set of metabolites considered in the joint regression (LASSO or another approach) analysis may also result in identification of different biomarkers. While this may indicate a potential limitation of biomarker development using untargeted metabolomics, it also suggests opportunities to identify metabolite associations and develop biomarkers for subphenotypes of a disease, e.g., for SDB that is driven by specific endotypes. It also highlights the importance of replication analyses and for the accumulation of evidence across multiple studies, followed by rigorous targeted studies.”

We also alluded to it in the limitation section of the discussion, page 35 line 22 (albeit indirectly):

“In this work we did not attempt to replicate the observed associations in MESA, because of the limited number of metabolites common in both HCHS/SOL and MESA.”

Following your comment, we also decided to report the association of the SDB PCs-associated metabolites with the individual SDB phenotypes contributing to the PCs. We reported this in the methods section, page 11, line 15:

“To further understand the SMA results in relation to the original SDB phenotypes, we also conducted the SMA between individual metabolites with replicated associations with the SDB PCs and the 7 individual SDB phenotypes (rank normalized) individually”.

Results section, page 24 line 12:

“In a secondary analysis, we compared the associations between SDB PCs-associated metabolites (FDR p-value based on batch 1 analysis for each SDB PC1 or PC2; metabolites reported in Figure 3) and the 7 individual SDB phenotypes comprising the SDB PCs. The four SDB PC2-associated metabolites had the largest evidence of association with average oxyhemoglobin saturation during sleep and with REI3. SDB PC1-associated metabolites were marginally associated with multiple SDB phenotypes, i.e., did not seem to strongly reflect associations with any specific individual SDB phenotype. Three metabolites, 1-stearoyl-2-arachidonoyl-GPC (18:0/20:4), 2-linoleoylglycerol (18:2) and glucuronide of C10H18O2 (8), had low evidence of association with all SDB phenotypes when evaluated individually (p-value>0.01; **Supplementary Figure S6**), highlighting the contribution of the PCA approach.”

2) The metabolomics workflow is lacking some information. What was the process used for identifying/annotating metabolite names? Were there any unknown metabolites? Can you please provide the Metabolomics Standard Initiative numbers for level of confidence in the metabolite annotations? Please also provide a standard metabolite identification number such as HMDB, PubChem, Lipid Maps, etc. This information will be helpful for future replication studies. For example you identify a sphingomyelin as sphingomyelin(d18:2/23:0,d18:1/23:1,d17:1/24:1)...does this indicate a sphingomyelin with 3 potential different combinations of fatty acids?

Response: We appreciate the reviewer’s comment regarding the metabolomics workflow. The identification and annotation of metabolite names and other biochemical information was done by Metabolon, Inc. In Supplementary Figure S1, we explained how the metabolites and samples were selected, including, unknown metabolites were excluded from the study. We also added Supplementary Table S3 which provides additional annotation, including HMDB, kegg, and PubChem IDs for the metabolites identified in the SMA analysis. We also provide Supplementary Note 2, methods applied by Metabolon, Inc. This note is referenced in the methods section.

3) Please clarify if any of the participants were undergoing treatment for SDB at time of sample collection or if such treatment could have influenced the analyses focused on incidence of diabetes and hypertension?

Response: We appreciate reviewer’s question regarding SDB treatment among the study population. Less than 1% of the study participants reported “ever been prescribed CPAP, BIPAP or oral device” (batch 1: 29 out of 3299, batch 2: 17 out of 1522). Given the small percentage and the ambiguity of the self-report (i.e., “prescribed” rather than “received/adhered to” treatment) and the low level of CPAP adherence in the population, we don’t think it would have

substantial impact on the study findings. Still, this is an important point and we added the following to the discussion section, page 36, line 17:

“Another limitation of this analysis is that some of the participants may have been treated for OSA. Less than 1% of the study participants (29 out of 3299 batch 1, and 17 out of 1522 batch 2 participants) reported “ever prescribed CPAP, BIPAP or oral device treatment”. Still, it is possible that some participants started treatment following the results from their home sleep apnea testing at the baseline examination. While it is unknown if these participants were actually treated or adhered to treatment, it is possible that OSA treatment in some study participants may have weakened the observed associations with incident cardiometabolic outcomes.”

4) Related, did you collect information on medications participants were taking, especially lipid lowering, anti-hypertensive, and glucose lowering medications? Could these potential medications influence the identified metabolites and incidence of diabetes and hypertension?

Response: Good questions. We do have information about medications and as metabolites are influenced by medications it is possible that medication use had some influence on the results. To address this comment, we added a secondary analysis excluding participants who reported any antihypertensive or antidiabetic medication use at baseline (remaining batch 1 participants: n=2581, remaining batch 2 participants: n=1030) for both LASSO linear regression and the association analysis with incident outcomes (Supplementary Table S16). Note that in this analysis we also addressed another comment about potential effects of sample overlap in the development of the PCs and of subsequent analysis by limiting the PCA to batch 1, non-medication users.

We first report this in the method section, page 13, line 1:

“As a sensitivity analysis, we assessed the robustness of the developed MRSs by limiting potential impacts of sample overlap and of medication use. Thus, we applied LASSO regression to develop MRS using a new set of SDB PCs developed exclusively among the batch 1 participants who were not taking any antihypertensives or antidiabetics at baseline.

Next in the method section, page 14, line 18:

“In a sensitivity analysis, we estimated the associations between the SDB PCs MRS developed in a process (both PCA and LASSO) involving only batch 1 participants who were not taking antihypertensive or antidiabetic at baseline and incident cardiometabolic outcomes within batch 2 participants who also did not take antihypertensive or antidiabetic at baseline.”

Result section, page 30, line 2:

“In the sensitivity analysis, we considered SDB PC MRSs developed via an analytic pipeline including both PCA and LASSO analysis restricted to batch 1 participants who were not taking any antihypertensive or antidiabetic medications at baseline. We refer to these MRSs as b1-SDB PC1 and b1-SDB PC2. These MRSs were each associated with increased risk for incident diabetes among batch 2 participants, while b1-SDB PC1 MRS but not b1-SDB PC2 was associated with increased risk of incident hypertension (associations were in both regression models 1 and 2; **Supplementary Table S15**). In comparison, SDB PC2 MRS from the main analysis was associated with incident hypertension. Otherwise, the effect estimates of b1-SDB PC MRSs were generally stronger than the SDB PC MRSs developed in the primary analysis. Lack of observed association (of b1-SDB PC2 MRS with incident hypertension) and differences in magnitude of effect sizes between main and sensitivity analyses should be interpreted with caution due to lower sample sizes in sensitivity analysis and the differences in sample representing healthier individuals.”

5) Please consider commenting on the fact that the PCA analysis of the SDB outcomes were conducted on all participants together, including the training and replication datasets. This means the PCS analyses already identified participants in both the training and replication datasets that have similar (or group together) clinical SDB features which may bias the metabolomics analyses to have a higher probability of replicating associations with the PC1 and PC2. If the PCA is conducted on the training set first, does it replicate in your replication dataset? This is an important consideration to determine if the PCA analysis, which all subsequent analyses are based on, is generalizable outside your study participants.

Response: Thank you for this comment. To address it, we performed a sensitivity analysis where we applied the entire analytic pipeline for MRS envelopment, starting from PCA and followed by LASSO linear regression, in batch 1 participants only. We then constructed both the SDB PCs and the SDB PC MRSs in batch 2 based on the output from the batch 1 analysis, then we assessed the associations with incident outcomes among batch 2 participants only. To simplify communication, we name the products of this sensitivity analysis with “b1”, i.e., “b1-SDB PCs” and “b1-SDB PC MRSs”. We assessed (1) the association between b1-SDB PCs and b1-SDB PC MRSs among batch 2 participants (Supplementary Table S10; they were highly associated), and (2) the association between b1-SDB PC MRSs with incident outcomes among batch 2 participants (Supplementary Table S15; and compared with Supplementary Table S12; strong association).

We report these analyses first in the methods section, first in page 10, line 1:

“As a sensitivity analysis, we also implemented PCA among the batch 1 participants, and derived SDB PCs for the batch 2 participants based on the loadings from batch 1, with SDB phenotypes rank normalized within each batch separately”.

Next in page 13, line 1:

“As a sensitivity analysis, we assessed the robustness of the developed MRSs by limiting potential impacts of sample overlap and of medication use. Thus, we applied LASSO

regression to develop MRS using a new set of SDB PCs developed exclusively among the batch 1 participants who were not taking any antihypertensives or antidiabetics at baseline.”

And in finally in the methods section in the context of association testing with incident outcomes, page 14, line 18:

“In a sensitivity analysis, we estimated the associations between the SDB PCs MRS developed in a process (both PCA and LASSO) involving only batch 1 participants who were not taking antihypertensive or antidiabetic at baseline and incident cardiometabolic outcomes within batch 2 participants who also did not take antihypertensive or antidiabetic at baseline.”

Result section, page 25, Line 20 (reporting that the association of the MRSs with their PC phenotypes existing in batch 2):

“In the sensitivity analysis, we constructed SDB PC MRS in a process restricted exclusively to batch 1 participants (including both PCA and MRS derivation). We then constructed the resulting b1-SDB PCs and b1-SDB PC MRSs among batch 2 participants. In batch 2, b1-SDB PCs and their corresponding b1-SDB PC MRSs were highly associated (Supplementary Table S10).”

And finally reporting that the b1-SDB PC MRSs were associated with incident outcomes in batch 2 participants: (page 30, line 2):

“In the sensitivity analysis, we considered b1-SDB PC MRSs developed via an analytic pipeline including both PCA and LASSO analysis restricted to batch 1 participants who were not taking any antihypertensive or antidiabetic medications at baseline. We refer to these MRSs as b1-SDB PC1 and b1-SDB PC2 MRSs. They were both associated with increased risk for incident diabetes among batch 2 participants, while only b1-SDB PC1 MRS was associated with increased risk of incident hypertension (associations were in both regression models 1 and 2; **Supplementary Table S15**). In comparison, SDB PC2 MRS from the main analysis was associated with incident hypertension. Otherwise, the effect estimates of b1-SDB PC MRSs were generally stronger than the SDB PC MRSs developed in the primary analysis. Lack of observed association (of b1-SDB PC2 MRS with incident hypertension) and differences in magnitude of effect sizes between main and sensitivity analyses should be interpreted with caution due to lower sample sizes in sensitivity analysis and the differences in sample representing healthier individuals”.

6) PC1 explains ~65% of variance whereas PC2 explains ~14% of the variance, does this imply the results and metabolites linked to PC1 have a higher probability of being more strongly related to SDB?

Response: Yes, given the percentage of variance explained by SDB PC1 versus PC2, we think it's fair to assume SDB PC1 is more strongly associated with the 7 SDB phenotypes compared to SDB PC2.

To clarify the difference in variance explained by the PCs, we added this to the results (new text in bold), page 18 line 5:

“The first two principal components of the SDB measures accounted for 79.8% of the total variance (**SDB PC1: 65%, SDB PC2: 14.5%**; Supplementary Figure S2)”

7) Model 1 includes body weight and BMI as covariates, do these variables represent independent characteristics or are they correlated with each other?

Response: Thank you for the question. We apologize for the ambiguity in the text, we only adjusted for BMI in the model, and not for body weight. We have revised the text to reflect that.

Method section, page 10, line 12:

“All analyses used up to three conceptual models. Model 1 (i.e., base model) adjusted for demographic variables, including age, sex, field center, Hispanic/Latino background (Mexican, Puerto Rican, Cuban, Central American, Dominican, and South American and other/multi), and body mass index (BMI).”

8) As stated on page 10 lines 214-219 "Model 3 is a lifestyle and comorbidity model that is adjusted for Model 2 variables in addition to indicators for diabetes mellitus and hypertension, and continuous measures of fasting insulin, fasting glucose, HOMA IR, HDL, LDL, total cholesterol, triglycerides, systolic blood pressure and diastolic blood pressure.". The continuous measures appear to be indicators for diabetes mellitus and hypertension, but as stated it seems there are other indicators for diabetes mellitus and hypertension in the model? Please clarify precisely what variables are in model 3.

Response: Thank you for the comment, we have clarified that the diabetes mellitus and hypertension indicators, although reflecting information from continuous measures such as fasting glucose, blood pressure, etc., also take into account additional information such as medication use, in the manuscript. The revised text is in the method section, page 10, line 19:

“Model 3 is a lifestyle and comorbidity model that is adjusted for Model 2 variables in addition and continuous measures of fasting insulin, fasting glucose, HOMA-IR, HDL, LDL, total cholesterol, triglycerides, systolic blood pressure, diastolic blood pressure, and also to indicators for diabetes mellitus and hypertension, which encapsulate additional information about medication use, not captured by continuous measures.”

9) Related, model 3 is minimally used in the analyses but could be highly informative. For example, do the variable metabolite biomarkers still associate with incidence of diabetes or hypertension when model 3 is used? This is important to understand if adding the metabolite biomarkers to traditional diabetes and hypertension clinical risk factors improves model

performance or risk prediction or if the traditional clinical risk factors already account for the variability/risk explained by the new biomarker models? This is especially important because your analyses with incidence of diabetes and hypertension used the training and replication datasets together and there was no validation assessment for these analyses, decreasing the rigor of the findings.

Response: We thank the reviewer for the comment. We would like to first clarify that model 3 covariates (adjusting for demographic, lifestyle, and comorbidities) were used only in the single metabolite analysis, and not included in the association analysis for the incident outcomes, since our goals were to understand SDB phenotypes and incident cardiometabolic outcomes, rather than developing a new risk prediction model for incident hypertension or diabetes.

In order to address the comment on the use of the combined dataset of batch 1 (training/discovery in SMA) and batch 2 (replication/validation in SMA), we now added a sensitivity analysis that only performed the entire MRS derivation analytic pipeline (PCA + MRS development) in batch 1 participants, further restricting to individuals not using medications, followed by MRS association analyses in batch 2. We describe this in the response to your comment number 5.

To further clarify the gap that MRS currently address and our expectation for future work, we revised the final sentence of the discussion to reduce the focus on prediction (page 38, line 13). It now reads:

“These findings provide a strong basis for the use of metabolomics in studying SDB, including for clarifying and measuring risk for incident outcomes by different quantitative SDB phenotypes and dichotomous subtypes. Future evaluations are needed to study the use of MRSs for risk stratification, treatments response, and ultimately as biomarkers that guide diagnosis and treatment decisions.”

10) In the replication analyses with one-sided p-values it appears no FDR correction was applied. Please confirm and justify why no FDR correction was needed/used for these analyses, do the findings change if a FDR correction is used?

Response: Thank you for the comment. We revised the analysis and we now apply FDR correction to the one-sided p-values.

In the method section, page 11, line 13:

“We computed one-sided p-values guided by the estimated directions of the associations in batch 1, and determined replication if the FDR corrected one-sided p-value was <0.05.”

Next in the result section, page 22, line 4 (updated text bolded):

“Among the 15 SDB PC1 metabolites, **two** metabolites, pregnanolone/allopregnanolone sulfate and glucuronide of C₁₀H₁₈O₂ (8), **had replicated associations (FDR-corrected one-sided p-value < 0.05)** in batch 2 in Model 1 analysis (Figure 4), and remained

associated (**pregnanolone/allopregnanolone sulfate: FDR-corrected one-sided p=.036 in model 2, FDR-corrected one-sided p=.039 in model 3; glucuronide of C10H18O2 (8): FDR-corrected one-sided p=.030 in model 2, FDR-corrected one-sided p=.032 in model 3**) with PC1 when adjusted for additional lifestyle and comorbidity covariates in batch 2. Three of the four metabolite associations with SDB PC2 in batch 1 replicated in batch 2 (**FDR-corrected one-sided p-value < 0.05**) in Model 1 and 2, all of which were sphingomyelin lipids - sphingomyelin(d18:2/24:2), sphingomyelin(d18:2/24:1,d18:1/24:2), and sphingomyelin(d18:2/23:0,d18:1/23:1, d17:1/24:1). Full results from the SMA sex-combined analysis are provided in Supplementary Table S4.”

11) Please clarify if metabolite concentrations in your models were relative concentrations or absolute concentrations and what implication this has for translating biomarkers to clinical applications.

Response: Metabolites in our analyses were represented as relative concentrations as provided by Metabolon platform. We clarify this in the text, methods section, page 8, line 2:

“Metabolite quantitation was relative, so that relationship between metabolite levels between metabolites and between individuals in the same sample are consistent, but the specific metabolite concentration for a given person and a given metabolite values may be different from those obtained by an absolute quantitation method. Detailed methods of metabolite assaying by Metabolon, Inc, are provided in Supplementary Note 2.”

12) SDB PC2 MRS was associated with incident diabetes but SDB PC2 itself was not. This is intriguing as SDB PC2 MRS was designed to represent or capture SDB PC2. These divergent associations with incident diabetes are somewhat hard to interpret then, especially without a replication/validation set in these analyses. Could this be potentially related to PC2 only explaining a relatively small proportion of the SDB measures (~14%)? Does these divergent findings limit the potential clinical utility of these biomarkers?

Response: We appreciate reviewer’s thoughtful comment. We now added a sensitivity analysis in which the MRS association analysis with incident outcomes is in batch 2 only while all the preceding derivation is in batch 1 only (as described in response to comments number 4 and 5). It’s worth noting that in the replication analysis, the associations between “b1-SDB PC2” and “b1-SDB PC2 MRS” (the variables derived solely using batch 1 participants) were statistically significant among batch 2 participants, despite the fact that neither measure was developed using any of the batch 2 participants. Moreover, the b1-SDB PC2-MRS showed a stronger effect estimate for the association with incident diabetes than the original SDB PC2 MRS (Supplementary Table S15).

It's also worth noting that SDB PCs were not designed to predict or associate with any specific outcome. However, given that OSA MRS is a better predictor than REI3 for the incident outcomes, it is possible that the MRSs more accurately capture the underlying metabolic

environment. We think that the variability in single-night polysomnography (PMID: 12489889) may contribute to the lower predictive power in SDB PCs for incident outcomes. In contrast, metabolomic profiles have shown relatively high short-term within-person stability in large cohorts (PMID: 23897902, 26274920). It is important to further assess and compare stability of both physiological and metabolomics biomarkers in future work.

We further addressed this comment in the discussion paragraph about MRSs, page 34, line 20:

“While we are unable to verify this hypothesis using existing data, one explanation for this discrepancy is that there is variability in phenotypes generated from single-night polygraphy, reducing the predictive ability of derived physiological traits. In contrast, metabolomic profiles constructed as MRS may be more stable. In addition, MRS may better describe the metabolomic environment compared to SDB phenotypes that focus on breathing-related variables.”

13) Supplementary Table S12 is hard to read/interpret as uploaded for the reviewers, a minor comment.

Response: Thank you for the comment. We have reformatted the table and we believe it is clearer (it is now Supplementary Table S14). The entire excel file is uploaded and we hope that reviewers can see it as such rather than as a pdf.

14) Given the cross-sectional study design and lack of replication for the analyses with incident diabetes and hypertension, please more explicitly state that the direction of causality cannot be inferred in current analyses. It is plausible that more severe SDB could be driving the changes in metabolites as opposed to changes in metabolites driving more severe SDB. This is an important consideration for your statement in Discussion on page 29 lines 500-503: "If true, these data point to the possibility that different classes of steroids of different origins may be involved in the development of SDB, and its association with incident hypertension and diabetes mellitus.".

Response: we completely agree. We edited the limitation paragraph in the discussion, which now includes the statement (page 36, line 13):

“Given the observational nature of the study, and the cross-sectional association between the SDB PCs and metabolites, we cannot draw causal inferences. Thus, we cannot determine whether metabolite levels contribute to more severe SDB, or if SDB severity causes metabolomic changes, or if there are common mechanisms that influence both SDB and metabolites.”

Regarding the sentence about incident hypertension and diabetes, we think that it is phrased in a way that does not over-interpret the results.

REVIEWER COMMENTS

Reviewer #1 (Remarks to the Author):

Thank you for the revisions. The manuscript is strengthened by the large prospective cohort, replication analysis, and multiple models that include a large number of demographic, lifestyle, and clinical covariates. There are limitations in keeping with the study design. First, the study uses a nontargeted approach to identify metabolites, so it serves as exploratory analysis though the Benjamini-Hochberg method was used for false discovery rate.

Additionally, the principal component composite measures are more difficult to interpret but were associated with incident outcomes, unlike single physiologic variables. Can you please add a succinct summary of how you interpret the identified metabolites in relation to PC1 versus PC2 to the abstract and discussion?

As has been mentioned, the generation of PCs and identification of metabolites was cross-sectional. Only Model 3 for the identification of metabolites adjusts for HTN/DM, and none of the identified metabolites were significantly associated with PC2 in Model 3 (so the metabolites may not be significantly predictive of SDB phenotypes independent of DM/HTN). Since causality cannot be determined with cross-sectional analysis, it is possible that the metabolites identified are markers of DM/HTN more generally rather than markers of incident DM/HTN in patients with SDB phenotypes. Is there any known data about the association between the identified metabolites like progesterone steroids sulfate metabolites and sphingomyelins with DM/HTN?

As minor points, could the definition of OSA in this cohort be included in subtext for supplemental table S1? Also I would include the limitation that measurement of metabolites were relative, which may limit the utility in clinical applications.

Reviewer #2 (Remarks to the Author):

Thank you for addressing the reviewer comments. I have no further comments on this manuscript. Please note there are a few typos to correct throughout the paper.

Response to review of NCOMMS-23-30340A “Metabolomic Profiles of Sleep-Disordered Breathing are Associated with Hypertension and Diabetes Mellitus Development: the HCHS/SOL”

REVIEWER COMMENTS

Reviewer #1 (Remarks to the Author):

Thank you for the revisions. The manuscript is strengthened by the large prospective cohort, replication analysis, and multiple models that include a large number of demographic, lifestyle, and clinical covariates. There are limitations in keeping with the study design. First, the study uses a nontargeted approach to identify metabolites, so it serves as exploratory analysis though the Benjamini-Hochberg method was used for false discovery rate.

Response: Thank you for your review. We would like to note that discovery studies in datasets of large sample sizes have been extremely effective for their purposes. Targeted approaches are limited in that they rely on knowledge or assumptions in the sense of “looking under the streetlight”. There are advantages to both study designs.

Additionally, the principal component composite measures are more difficult to interpret but were associated with incident outcomes, unlike single physiologic variables. Can you please add a succinct summary of how you interpret the identified metabolites in relation to PC1 versus PC2 to the abstract and discussion?

Response: There were 5 metabolite associations identified and replicated with either PC1 or PC2 in cross-sectional analysis adjusted for BMI, demographic and lifestyle covariates. In the discussion, we already described the associations of specific metabolites with PC1 and with PC2 and we think that it is sufficient discussion. In the abstract, we added the following sentence to address your comment:

“SDB PC1, which characterizes an SDB phenotype of frequent respiratory events common in older and male adults, was associated with pregnanolone and progesterone-related sulfated metabolites. SDB PC2, which reflects an SDB phenotype characterized by short respiratory event length, self-reported restless sleep, and is enriched in young adults, was associated with levels of multiple sphingomyelins.”

We think that this sentence provides additional, useful synthesis, yet does not over-interpret. We wanted to avoid over-interpretation given limitations (e.g., not having all metabolites and steroids available).

As has been mentioned, the generation of PCs and identification of metabolites was cross-sectional. Only Model 3 for the identification of metabolites adjusts for HTN/DM, and none of the identified metabolites were significantly associated with PC2 in Model 3 (so the metabolites may not be significantly predictive of SDB phenotypes independent of DM/HTN). Since causality cannot be determined with cross-sectional analysis, it is possible that the metabolites identified are markers of DM/HTN more generally rather than markers of incident DM/HTN in patients with SDB phenotypes. Is there any known data about the association between the identified metabolites like progesterone steroids sulfate metabolites and sphingomyelins with DM/HTN?

Response: Thank you for suggesting this. The reviewer is correct to point out that discovery of single metabolite associations was limited to cross-sectional analysis, and that this may imply specific metabolites may not be unique to SDB but rather are associated with SDB comorbidities. It is interesting to observe that for SDB PC1 associated metabolites, the effect size estimates change only slightly between model 1 and 3, while the change in effect size estimates is more substantial for SDB PC2 associations (see supplementary table S2). Thus, we added the following paragraph to the discussion section (page 35, line):

“The associations between these sphingomyelins and SDB PC2 were no longer statistically significant (FDR-corrected $p > .05$) in analysis adjusted for comorbidities (blood pressure-related phenotypes, diabetes and glycemic phenotypes, and cholesterol and lipid measures). Since diabetes mellitus and hypertension are common comorbidities to SDB, it is possible that these PC2-associated metabolites may partly reflect metabolic state related to these diseases, rather than being specific to SDB. Sphingolipids have been shown to mediate loss of insulin sensitivity, and to promote diabetic proinflammatory state, although the roles of specific sphingolipid species and pathways remain obscure⁶⁶.”

As minor points, could the definition of OSA in this cohort be included in subtext for supplemental table S1?

Response: We thank the reviewer for the suggestion, and we have added the OSA definition to the supplemental table S1.

Also I would include the limitation that measurement of metabolites were relative, which may limit the utility in clinical applications.

Response: This is already discussed in the manuscript (Page 8, line 2):

“Metabolite quantitation was relative, so that relationship between metabolite levels between metabolites and between individuals in the same sample are consistent, but the specific metabolite concentration for a given person and a given metabolite values may be different from those obtained by an absolute quantitation method.”

Reviewer #2 (Remarks to the Author):

Thank you for addressing the reviewer comments. I have no further comments on this manuscript. Please note there are a few typos to correct throughout the paper.

Response: Thank you. We reviewed the paper and fixed some types.

REVIEWERS' COMMENTS

Reviewer #1 (Remarks to the Author):

Thank you for the responses to the reviews. I have no further comments.